# ROBUST REINFORCEMENT LEARNING USING ADVERSARIAL POPULATIONS

## ABSTRACT

Reinforcement Learning (RL) is an effective tool for controller design but can struggle with issues of robustness, failing catastrophically when the underlying system dynamics are perturbed. The Robust RL formulation tackles this by adding worst-case adversarial noise to the dynamics and constructing the noise distribution as the solution to a zero-sum minimax game. However, existing work on learning solutions to the Robust RL formulation has primarily focused on training a single RL agent against a single adversary. In this work, we demonstrate that using a single adversary does not consistently yield robustness to dynamics variations under standard parametrizations of the adversary; the resulting policy is highly exploitable by new adversaries. We propose a population-based augmentation to the Robust RL formulation in which we randomly initialize a population of adversaries and sample from the population uniformly during training. We empirically validate across robotics benchmarks that the use of an adversarial population results in a less exploitable, more robust policy. Finally, we demonstrate that this approach provides comparable robustness and generalization as domain randomization on these benchmarks while avoiding a ubiquitous domain randomization failure mode.

## 1 INTRODUCTION

Developing controllers that work effectively across a wide range of potential deployment environments is one of the core challenges in engineering. The complexity of the physical world means that the models used to design controllers are often inaccurate. Optimization based control design approaches, such as reinforcement learning (RL), have no notion of model inaccuracy and can lead to controllers that fail catastrophically under mismatch. In this work, we aim to demonstrate an effective method for training reinforcement learning policies that are robust to model inaccuracy by designing controllers that are effective in the presence of worst-case adversarial noise in the dynamics.

An easily automated approach to inducing robustness is to formulate the problem as a zero-sum game and learn an adversary that perturbs the transition dynamics (Tessler et al., 2019; Kamalaruban et al., 2020; Pinto et al., 2017). If a global Nash equilibrium of this problem is found, then that equilibrium provides a lower bound on the performance of the policy under some bounded set of perturbations. Besides the benefit of removing user design once the perturbation mechanism is specified, this approach is maximally conservative, which is useful for safety critical applications.

However, the literature on learning an adversary predominantly uses a single, stochastic adversary. This raises a puzzling question: the zero-sum game does not necessarily have any pure Nash equilibria (see Appendix C in Tessler et al. (2019)) but the existing robust RL literature mostly appears to attempt to solve for pure Nash equilibria. That is, the most general form of the minimax problem searches over distributions of adversary and agent policies, however, this problem is approximated in the literature by a search for a single agent-adversary pair. We contend that this reduction to a single adversary approach can sometimes fail to result in improved robustness under standard parametrizations of the adversary policy.

The following example provides some intuition for why using a single adversary can decrease robustness. Consider a robot trying to learn to walk east-wards while an adversary outputs a force representing wind coming from the north or the south. For a fixed, deterministic adversary the agent knows that the wind will come from either south or north and can simply apply a counteracting force at each state. Once the adversary is removed, the robot will still apply the compensatory forces and

possibly become unstable. Stochastic Gaussian policies (ubiquitous in continuous control) offer little improvement: they cannot represent multi-modal perturbations. Under these standard policy parametrizations, we cannot use an adversary to endow the agent with a prior that a strong wind could persistently blow either north or south. This leaves the agent exploitable to this class of perturbations.

The use of a single adversary in the robustness literature is in contrast to the multi-player game literature. In multi-player games, large sets of adversaries are used to ensure that an agent cannot easily be exploited (Vinyals et al., 2019; Czarnecki et al., 2020; Brown & Sandholm, 2019). Drawing inspiration from this literature, we introduce **RAP** (Robustness via Adversary Populations): a randomly initialized population of adversaries that we sample from at each rollout and train alongside the agent. Returning to our example of a robot perturbed by wind, if the robot learns to cancel the north wind effectively, then that opens a niche for an adversary to exploit by applying forces in another direction. With a population, we can endow the robot with the prior that a strong wind could come from either direction and that it must walk carefully to avoid being toppled over.

Our contributions are as follows:

- Using a set of continuous robotics control tasks, we provide evidence that a single adversary does not have a consistent positive impact on the robustness of an RL policy while the use of an adversary population provides improved robustness across all considered examples.
- We investigate the source of the robustness and show that the single adversary policy is exploitable by new adversaries whereas policies trained with RAP are robust to new adversaries.
- We demonstrate that adversary populations provide comparable robustness to domain randomization while avoiding potential failure modes of domain randomization.

## 2 RELATED WORK

This work builds upon robust control (Zhou & Doyle, 1998), a branch of control theory focused on finding optimal controllers under worst-case perturbations of the system dynamics. The Robust Markov Decision Process (R-MDP) formulation extends this worst-case model uncertainty to uncertainty sets on the transition dynamics of an MDP and demonstrates that computationally tractable solutions exist for small, tabular MDPs (Nilim & El Ghaoui, 2005; Lim et al., 2013). For larger or continuous MDPs, one successful approach has been to use function approximation to compute approximate solutions to the R-MDP problem (Tamar et al., 2014).

One prominent variant of the R-MDP literature is to interpret the perturbations as an adversary and attempt to learn the distribution of the perturbation under a minimax objective. Two variants of this idea that tie in closely to our work are Robust Adversarial Reinforcement Learning (RARL)(Pinto et al., 2017) and Noisy Robust Markov Decision Processes (NR-MDP) (Tessler et al., 2019) which differ in how they parametrize the adversaries: RARL picks out specific robot joints that the adversary acts on while NR-MDP adds the adversary action to the agent action. Both of these works attempt to find an equilibrium of the minimax objective using a single adversary; in contrast our work uses a large set of adversaries and shows improved robustness relative to a single adversary.

A strong alternative to the minimax objective, domain randomization, asks a designer to explicitly define a distribution over environments that the agent should be robust to. For example, (Peng et al., 2018) varies simulator parameters to train a robot to robustly push a puck to a target location in the real world; (Antonova et al., 2017) adds noise to friction and actions to transfer an object pivoting policy directly from simulation to a Baxter robot. Additionally, domain randomization has been successfully used to build accurate object detectors solely from simulated data (Tobin et al., 2017) and to zero-shot transfer a quadcopter flight policy from simulation (Sadeghi & Levine, 2016).

The use of population based training is a standard technique in multi-agent settings. Alphastar, the grandmaster-level Starcraft bot, uses a population of "exploiter" agents that fine-tune against the bot to prevent it from developing exploitable strategies (Vinyals et al., 2019). (Czarnecki et al., 2020) establishes a set of sufficient geometric conditions on games under which the use of multiple adversaries will ensure gradual improvement in the strength of the agent policy. They empirically demonstrate that learning in games can often fail to converge without populations. Finally, Active Domain Randomization (Mehta et al., 2019) is a very close approach to ours, as they use a population

of adversaries to select domain randomization parameters whereas we use a population of adversaries to directly perturb the agent actions. However, they explicitly induce diversity using a repulsive term and use a discriminator to generate the reward.

# 3 BACKGROUND

In this work we use the framework of a multi-agent, finite-horizon, discounted, Markov Decision Process (MDP) (Puterman, 1990) defined by a tuple $\langle A_{\text{agent}} \times A_{\text{adversary}}, S, \mathcal{T}, r, \gamma \rangle$. Here $A_{\text{agent}}$ is the set of actions for the agent, $A_{\text{adversary}}$ is the set of actions for the adversary, $S$ is a set of states, $\mathcal{T} : A_{\text{agent}} \times A_{\text{adversary}} \times S \to \Delta(S)$ is a transition function, $r : A_{\text{agent}} \times A_{\text{adversary}} \times S \to \mathbb{R}$ is a reward function and $\gamma$ is a discount factor. $S$ is shared between the adversaries as they share a state-space with the agent. The goal for a given MDP is to find a policy $\pi_\theta$ parametrized by $\theta$ that maximizes the expected cumulative discounted reward $J^\theta = \mathbb{E}\left[\sum_{t=0}^{T} \gamma^t r(s_t, a_t) | \pi_\theta\right]$. The conditional in this expression is a short-hand to indicate that the actions in the MDP are sampled via $a_t \sim \pi_\theta(s_t, a_{t-1})$. We denote the agent policy parametrized by weights $\theta$ as $\pi_\theta$ and the policy of adversary $i$ as $\bar{\pi}_{\phi_i}$. Actions sampled from the adversary policy $\bar{\pi}_{\phi_i}$ will be written as $\bar{a}_t^i$. We use $\xi$ to denote the parametrization of the system dynamics (e.g. different values of friction, mass, wind, etc.) and the system dynamics for a given state and action as $s_{t+1} \sim f_\xi(s_t, a_t)$.

## 3.1 BASELINES

Here we outline prior work and the approaches that will be compared with RAP. Our baselines consist of a single adversary and domain randomization.

### 3.1.1 SINGLE MINIMAX ADVERSARY

Our adversary formulation uses the *Noisy Action Robust MDP* (Tessler et al., 2019) in which the adversary adds its actions onto the agent actions. The objective is

$$\max_\theta \mathbb{E}\left[\sum_{t=0}^{T} \gamma^t r(s_t, a_t + \alpha \bar{a}_t) | \pi_\theta, \ \bar{\pi}_\phi\right]$$
$$\min_\phi \mathbb{E}\left[\sum_{t=0}^{T} \gamma^t r(s_t, a_t + \alpha \bar{a}_t) | \pi_\theta, \ \bar{\pi}_\phi\right]$$

(1)

where $\alpha$ is a hyperparameter controlling the adversary strength. This is a game in which the adversary and agent play simultaneously. We note an important restriction inherent to this adversarial model. Since the adversary is only able to attack the agent through the actions, there is a restricted class of dynamical systems that it can represent; this set of dynamical systems may not necessarily align with the set of dynamical systems that the agent may be tested in. This is a restriction caused by the choice of adversarial perturbation and could be alleviated by using different adversarial parametrizations e.g. perturbing the transition function directly.

### 3.1.2 DYNAMICS RANDOMIZATION

Domain randomization is the setting in which the user specifies a set of environments which the agent should be robust to. This allows the user to directly encode knowledge about the likely deviations between training and testing domains. For example, the user may believe that friction is hard to measure precisely and wants to ensure that their agent is robust to variations in friction; they then specify that the agent will be trained with a wide range of possible friction values. We use $\xi$ to denote some vector that parametrizes the set of training environments (e.g. friction, masses, system dynamics, etc.). We denote the domain over which $\xi$ is drawn from as $\Xi$ and use $\mathcal{P}(\Xi)$ to denote

some probability distribution over $\xi$. The domain randomization objective is

$$\max_\theta \mathbb{E}_{\xi \sim \mathcal{P}(\Xi)} \left[ \mathbb{E}_{s_{t+1} \sim f_\xi(s_t, a_t)} \left[ \sum_{t=0}^T \gamma^t r(s_t, a_t) | \pi_\theta \right] \right]$$
$$s_{t+1} \sim f_\xi(s_t, a_t)$$
$$a_t \sim \pi_\theta(s_t)$$

(2)

Here the goal is to find an agent that performs well on average across the distribution of training environment. Most commonly, and in this work, the parameters $\xi$ are sampled uniformly over $\Xi$.

# 4 RAP: ROBUSTNESS VIA ADVERSARY POPULATIONS

**RAP** extends the minimax objective with a population based approach. Instead of a single adversary, at each rollout we will sample uniformly from a population of adversaries. By using a population, the agent is forced to be robust to a wide variety of potential perturbations rather than a single perturbation. If the agent begins to overfit to any one adversary, this opens up a potential niche for another adversary to exploit. For problems with only one failure mode, we expect the adversaries to all come out identical to the minimax adversary, but as the number of failure modes increases the adversaries should begin to diversify to exploit the agent. To induce this diversity, we will rely on randomness in the gradient estimates and randomness in the initializations of the adversary networks rather than any explicit term that induces diversity.

Denoting $\bar{\pi}_{\phi_i}$ as the $i$-th adversary and $i \sim U(1, n)$ as the discrete uniform distribution defined on 1 through n, the objective becomes

$$\max_\theta \; \mathbb{E}_{i \sim U(1,n)} \left[ \sum_{t=0}^T \gamma^t r(s_t, a_t, \alpha \bar{a}_t^i) | \pi_\theta, \; \bar{\pi}_{\phi_i} \right]$$

$$\min_{\phi_i} \; \mathbb{E} \left[ \sum_{t=0}^T \gamma^t r(s_t, a_t, \alpha \bar{a}_t^i) | \pi_\theta, \; \bar{\pi}_{\phi_i} \right] \; \forall i = 1, \ldots, n$$

$$s_{t+1} \sim f(s_t, a_t + \alpha \bar{a}_t)$$

(3)

For a single adversary, this is equivalent to the *minimax adversary* described in Sec. 3.1.1. This is a game in which the adversary and agent play simultaneously.

We will optimize this objective by converting the problem into the equivalent zero-sum game. At the start of each rollout, we will sample an adversary index from the uniform distribution and collect a trajectory using the agent and the selected adversary. For notational simplicity, we assume the trajectory is of length T and that adversary $i$ will participate in $J_i$ total trajectories while, since the agent participates in every rollout, the agent will receive J total trajectories. We denote the j-th collected trajectory for the agent as $\tau_j = (s_0, a_0, r_0, s_1) \times \cdots \times (s_M, a_M, r_M, s_{M+1})$ and the associated trajectory for adversary $i$ as $\tau_j^i = (s_0, a_0, -r_0, s_1) \times \cdots \times (s_M, a_M, -r_M, s_M)$. Note that the adversary reward is simply the negative of the agent reward. We will use Proximal Policy Optimization (Schulman et al., 2017) (PPO) to update our policies. We caution that we have overloaded notation slightly here and for adversary $i$, $\tau_{j=1:J_i}^i$ refers only to the trajectories in which the adversary was selected: adversaries will only be updated using trajectories where they were active.

At the end of a training iteration, we update all our policies using gradient descent. The algorithm is summarized below:

---
**Algorithm 1:** Robustness via Adversary Populations

---
Initialize $\theta, \phi_1 \cdots \phi_n$ using Xavier initialization (Glorot & Bengio, 2010);
**while** *not converged* **do**
    **for** *rollout j=1...J* **do**
        sample adversary $i \sim U(1, n)$;
        run policies $\pi_\theta, \bar{\pi}_{\phi_i}$ in environment until termination;
        collect trajectories $\tau_j, \tau_j^i$
    **end**
    update $\theta, \phi_1 \cdots \phi_n$ using PPO (Schulman et al., 2017) and trajectories $\tau_j$ for $\theta$ and $\tau_j^i$ for each
      $\phi_i$;
**end**

---

## 5 EXPERIMENTS

In this section we present experiments on continuous control tasks from the OpenAI Gym Suite (Brockman et al., 2016; Todorov et al., 2012). We compare with the existing literature and evaluate the efficacy of a population of learned adversaries across a wide range of state and action space sizes. We investigate the following hypotheses:

H1. Agents are more likely to overfit to a single adversary than a population of adversaries, leaving them less robust on in-distribution tasks.

H2. Agents trained against a population of adversaries will generalize better, leading to improved performance on out-of-distribution tasks.

In-distribution tasks refer to the agent playing against perturbations that are in the training distribution: adversaries that add their actions onto the agent. However, the particular form of the adversary and their restricted perturbation magnitude means that there are many dynamical systems that they cannot represent (for example, significant variations of joint mass and friction). These tasks are denoted as out-of-distribution tasks. All of the tasks in the test set described in Sec. 5.1 are likely out-of-distribution tasks.

### 5.1 EXPERIMENTAL SETUP AND HYPERPARAMETER SELECTION

While we provide exact details of the hyperparameters in the Appendix, adversarial settings require additional complexity in hyperparameter selection. In the standard RL procedure, optimal hyperparameters are selected on the basis of maximum expected cumulative reward. However, if an agent playing against an adversary achieves a large cumulative reward, it is possible that the agent was simply playing against a weak adversary. Conversely, a low score does not necessarily indicate a strong adversary nor robustness: it could simply mean that we trained a weak agent.

To address this, we adopt a version of the train-validate-test split from supervised learning. We use the mean policy performance on a suite of validation tasks to select the hyperparameters, then we train the policy across ten seeds and report the resultant mean and standard deviation over twenty trajectories. Finally, we evaluate the seeds on a holdout test set of eight additional model-mismatch tasks. These tasks vary significantly in difficulty; for visual clarity we report the average across tasks in this paper and report the full breakdown across tasks in the Appendix.

We experiment with the Hopper, Ant, and Half Cheetah continuous control environments used in the original RARL paper Pinto et al. (2017); these are shown in Fig. 1. To generate the validation model mismatch, we pre-define ranges of mass and friction coefficients as follows: for Hopper, mass $\in [0.7, 1.3]$ and friction $\in [0.7, 1.3]$; Half Cheetah and Ant, mass $\in [0.5, 1.5]$ and friction $\in [0.1, 0.9]$. We scale the friction of every Mujoco geom and the mass of the torso with the same (respective) coefficients. We compare the robustness of agents trained via RAP against: 1) agents trained against a single adversary in a zero-sum game, 2) oracle agents trained using domain randomization, and 3) an agent trained only using PPO and no perturbation mechanism. To train the domain randomization

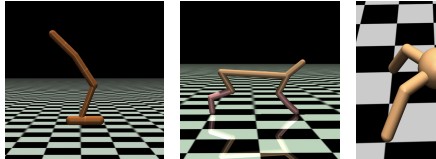

Figure 1: From left to right, the Hopper, Half-Cheetah, and Ant environments we use to test our algorithm.

oracle, at each rollout we uniformly sample a friction and mass coefficient from the validation set ranges. We then scale the friction of all geoms and the mass of the torso by their respective coefficients; this constitutes directly training on the validation set. To generate the test set of model mismatch, we take both the highest and lowest friction coefficients from the validation range and apply them to different combinations of individual geoms. For the exact selected combinations, please refer to the Appendix.

As further validation of the benefits of RAP, we include an additional set of experiments on a continuous control task, a gridworld maze search task, and a Bernoulli Bandit task in Appendix Sec. F. Finally, we note that both our agent and adversary networks are two layer-neural networks with 64 hidden units in each layer and a tanh nonlinearity.

## 6 RESULTS

**H1. In-Distribution Tasks: Analysis of Overfitting**
A globally minimax optimal adversary should be unexploitable and perform equally well against any adversary of equal strength. We investigate the optimality of our policy by asking whether the minimax agent is robust to swaps of adversaries from different training runs, i.e. different seeds. Fig. 2 shows the result of these swaps for the one adversary and three adversary case. The diagonal corresponds to playing against the adversaries the agent was trained with while every other square corresponds to playing against adversaries from a different seed. To simplify presentation, in the three adversary case, each square is the average performance against all the adversaries from that seed. We observe that the agent trained against three adversaries (top row right) is robust under swaps while the single adversary case is not (top row left). The agent trained against a single adversary is highly exploitable, as can be seen by its extremely sub-par performance against an adversary from any other seed. Since the adversaries off-diagonal are feasible adversaries, this suggests that we have found a poor local optimum of the objective.

In contrast, the three adversary case is generally robust regardless of which adversary it plays against, suggesting that the use of additional adversaries has made the agent more robust. One possible hypothesis for why this could be occurring is that the adversaries in the "3 adversary" case are somehow weaker than the adversaries in the "1 adversary" case. The middle row of the figure shows that it is not the case that the improved performance of the agent playing against the three adversaries is due to some weakness of the adversaries. If anything, the adversaries from the three adversary case are stronger as the agent trained against 1 adversary does extremely poorly playing against the three adversaries (left) whereas the agent trained against three adversaries still performs well when playing against the adversaries from the single-adversary runs. Finally, the bottom row investigates how an agent trained with domain randomization fairs against adversaries from either training regimes. In neither case is the domain randomization agent robust on these tasks.

**H2. Out-of-Distribution Tasks: Robustness and Generalization of Population Training**
Here we present the results from the validation and holdout test sets described in Section 5.1. We compare the performance of training with adversary populations of size three and five against vanilla PPO, the domain randomization oracle, and the single minimax adversary. We refer to domain randomization as an oracle as it is trained directly on the test distribution.

Fig.6 shows the average reward (the average of ten seeds across the validation or test sets respectively) for each environment. Table 1 gives the corresponding numerical values and the percent change of each policy from the baseline. Standard deviations are omitted on the test set due to wide variation in task difficulty; the individual tests that we aggregate here are reported in the Appendix with

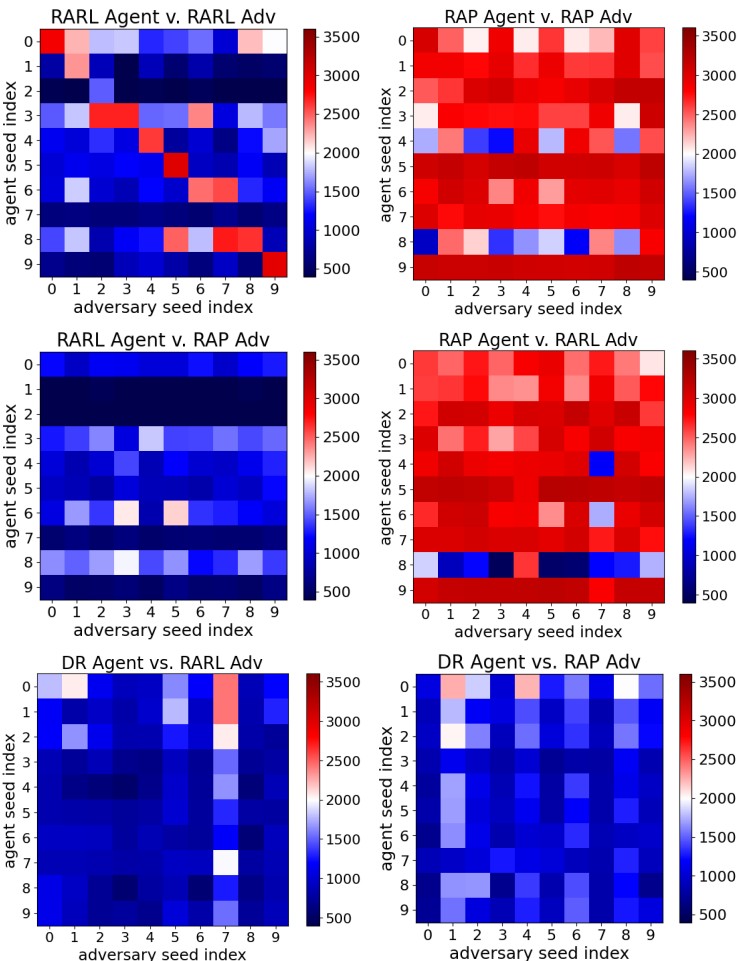

Figure 2: Top row: Average cumulative reward under swaps for one adversary training (left) and three-adversary training (right). Each square corresponds to 20 trials. In the three adversary case, each square is the average performance against the adversaries from that seed. Middle row: (Left) Playing the agent trained against 1 adversary against the adversaries from the three adversary case. (Right) Playing the agent trained against 3 adversaries against the adversaries from the one adversary case. Bottom row: (Left) Playing the DR agent against the adversaries from the three adversary case. (Right) Playing the DR agent against the adversaries from the one adversary case.

appropriate error bars. In all environments we achieve a higher reward across both the validation and holdout test set using RAP of size three and/or five when compared to the single minimax adversary case. These results from testing on new environments with altered dynamics supports hypothesis H2. that training with a population of adversaries leads to more robust policies than training with a single adversary in out-of-distribution tasks. Furthermore, while the performance is only comparable with the domain randomization oracle, the adversarial approach does not require prior engineering of appropriate randomizations. Furthermore, despite domain randomization being trained directly on these out-of-distribution tasks, domain randomization can have serious failure modes of domain randomization due to its formulation. A detailed analysis of this can be found in Appendix E.

For a more detailed comparison of robustness across the validation set, Fig. 4 shows heatmaps of the performance across all the mass, friction coefficient combinations. Here we highlight the heatmaps for Hopper and Half Cheetah for vanilla PPO, domain randomization oracle, single adversary, and best adversary population size. Additional heatmaps for other adversary population sizes and the Ant environment can be found in the Appendix. Note that Fig. 4 is an example of a case where a single adversary has negligible effect on or slightly reduces the performance of the resultant policy on the

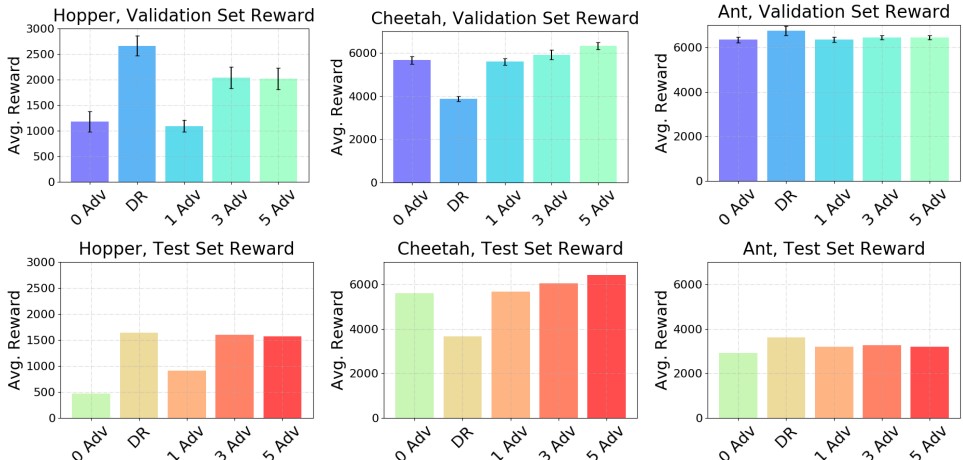

Figure 3: Average reward for Ant, Hopper, and Cheetah environments across ten seeds and across the validation set (top row) and across the holdout test set (bottom row). We compare vanilla PPO, the domain randomization oracle, and the minimax adversary against RAP of size three and five. Bars represent the mean and the arms represent the std. deviation. Both are computed over 20 rollouts for each test-set sample. The std. deviation for the test set are not reported here for visual clarity due to the large variation in holdout test difficulty.

| | Validation | | | | | Test | | | | |
|---|---|---|---|---|---|---|---|---|---|---|
| Ant | 0 Adv | DR | 1 Adv | 3 Adv | 5 Adv | 0 Adv | DR | 1 Adv | 3 Adv | 5 Adv |
| Mean Rew. | 6336 | *6743* | 6349 | 6432 | **6438** | 2908 | *3613* | 3206 | **3272** | 3203 |
| % Change | | *6.4* | 0.2 | 1.5 | **1.6** | | *24.3* | 10.2 | **12.5** | 10.2 |
| | Validation | | | | | Test | | | | |
| Hopper | 0 Adv | DR | 1 Adv | 3 Adv | 5 Adv | 0 Adv | DR | 1 Adv | 3 Adv | 5 Adv |
| Mean Rew. | 1182 | *2662* | 1094 | **2039** | 2021 | 472 | *1636* | 913 | **1598** | 1565 |
| % Change | | *125* | -7.4 | **72.6** | 71 | | *246* | 93.4 | **238** | 231 |
| | Validation | | | | | Test | | | | |
| Cheetah | 0 Adv | DR | 1 Adv | 3 Adv | 5 Adv | 0 Adv | DR | 1 Adv | 3 Adv | 5 Adv |
| Mean Rew. | 5659 | 3864 | 5593 | 5912 | *6323* | 5592 | 3656 | 5664 | 6046 | *6406* |
| % Change | | -32 | -1.2 | 4.5 | *11.7* | | -35 | 1.3 | 8.1 | *14.6* |

Table 1: Average reward and % change from vanilla PPO (0 Adv) for Ant, Hopper, and Cheetah environments across ten seeds and across the validation (left) or holdout test set (right). Across all environments, we see consistently higher robustness using RAP than the minimax adversary. Most robust adversarial approach is bolded as domain randomization is an oracle and outside the class of perturbations that our adversaries can construct, and best result overall is italicized.

validation set. This supports our hypothesis that a single adversary can actually lower the robustness of an agent.

# 7 CONCLUSIONS AND FUTURE WORK

In this work we demonstrate that the use of a single adversary to approximate the solution to a minimax problem does not consistently lead to improved robustness. We propose a solution through the use of multiple adversaries (RAP), and demonstrate that this provides robustness across a variety of robotics benchmarks. We also compare RAP with domain randomization and demonstrate that while DR can lead to a more robust policy, it requires careful parametrization of the domain we sample from to ensure robustness. RAP does not require this tuning, allowing for use in domains where appropriate tuning requires extensive prior knowledge or expertise.

There are several open questions stemming from this work. While we empirically demonstrate the effects of RAP, we do not have a compelling theoretical understanding of why multiple adversaries are helping. Perhaps RAP helps approximate a mixed Nash equilibrium as discussed in Sec. 1 or

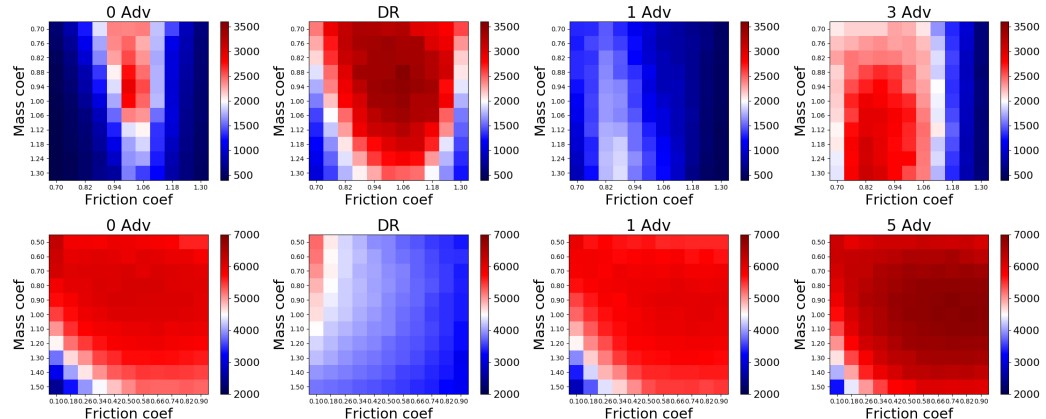

Figure 4: Average reward across ten seeds on each validation set parametrization – friction coefficient on the x-axis and mass coefficient on the y-axis. DR refers to domain randomization and X Adv is an agent trained against X adversaries. Top row is Hopper and bottom row is Half Cheetah.

perhaps population based training increases the likelihood that one of the adversaries is strong? Would the benefits of RAP disappear if a single adversary had the ability to represent mixed Nash?

There are some extensions of this work that we would like to pursue. We have looked at the robustness of our approach in simulated settings; future work will examine whether this robustness transfers to real-world settings. Additionally, our agents are currently memory-less and therefore cannot perform adversary identification; perhaps memory leads to a system-identification procedure that improves transfer performance. Our adversaries can also be viewed as forming a task distribution, allowing them to be used in continual learning approaches like MAML (Nagabandi et al., 2018) where domain randomization is frequently used to construct task distributions.

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

## A    FULL DESCRIPTION OF THE CONTINUOUS CONTROL MDPS

We use the Mujoco ant, cheetah, and hopper environments as a test of the efficacy of our strategy versus the 0 adversary, 1 adversary, and domain randomization baselines. We use the Noisy Action Robust MDP formulation Tessler et al. (2019) for our adversary parametrization. If the normal system dynamics are

$$s_{k+1} = s_k + f(s_k, a_k)\Delta t$$

the system dynamics under the adversary are

$$s_{k+1} = s_k + f(s_k, a_k + a_k^{\text{adv}})\Delta t$$

where $a_k^{\text{adv}}$ is the adversary action at time k.

The notion here is that the adversary action is passed through the dynamics function and represents some additional set of dynamics. It is standard to clip actions within some boundary but for the above reason, we clip the agent and adversary actions separately. Otherwise, an agent would be able to limit the effect of the adversary by always taking actions at the bounds of its clipping range. The agent is clipped between $[-1, 1]$ in the Hopper environment and the adversary is clipped between $[-.25, .25]$.

The MDP through which we train the agent policy is characterized by the following states, actions, and rewards:

- $s_t^{\text{agent}} = [o_t, a_t]$ where $o_t$ is an observation returned by the environment, and $a_t$ is the action taken by the agent.
- We use the standard rewards provided by the OpenAI Gym Mujoco environments at `https://github.com/openai/gym/tree/master/gym/envs/mujoco`. For the exact functions, please refer to the code at **ANONYMIZED**.
- $a_t^{\text{agent}} \in [a_{\min}, a_{\max}]^n$.

The MDP for adversary $i$ is the following:

- $s_t = s_t^{\text{agent}}$. The adversary sees the same states as the agent.
- The adversary reward is the negative of the agent reward.
- $a_t^{\text{adv}} \in \left[a_{\min}^{\text{adv}}, a_{\max}^{\text{adv}}\right]^n$.

For our domain randomization Hopper baseline, we use the following randomization: at each rollout, we scale the friction of all joints by a single value uniformly sampled from [0.7, 1.3]. We also randomly scale the mass of the 'torso' link by a single value sampled from [0.7, 1.3]. For Half-Cheetah and Ant the range for friction is [0.1, 0.9] and for mass the range is [0.5, 1.5].

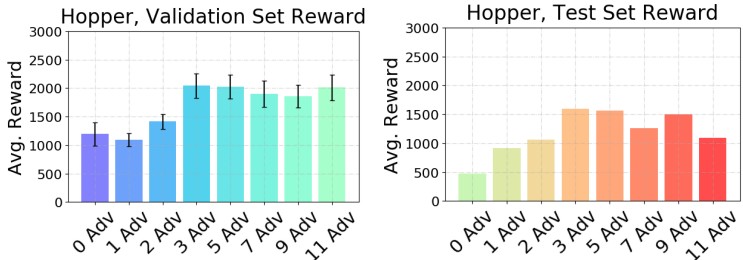

Figure 5: Average reward for Hopper across varying adversary number.

## B  INCREASING ADVERSARY POOL SIZE

We investigate whether **RAP** is robust to adversary number as this would be a useful property to minimize hyperparameter search. Here we hypothesize that while having more adversaries can represent a wider range of dynamics to learn to be robust to, we expect there to be diminishing returns due to the decreased batch size that each adversary receives (total number of environment steps is held constant across all training variations). We expect decreasing batch size to lead to worse agent policies since the batch will contain under-trained adversary policies. We cap the number of adversaries at eleven as our machines ran out of memory at this value. We run ten seeds for every adversary value and Fig. 5 shows the results for Hopper. Agent robustness on the test set increases monotonically up to three adversaries and roughly begins to decrease after that point. This suggests that a trade-off between adversary number and performance exists although we do not definitively show that diminishing batch sizes is the source of this trade-off. However, we observe in Fig. 6 that both three and five adversaries perform well across all studied Mujoco domains.

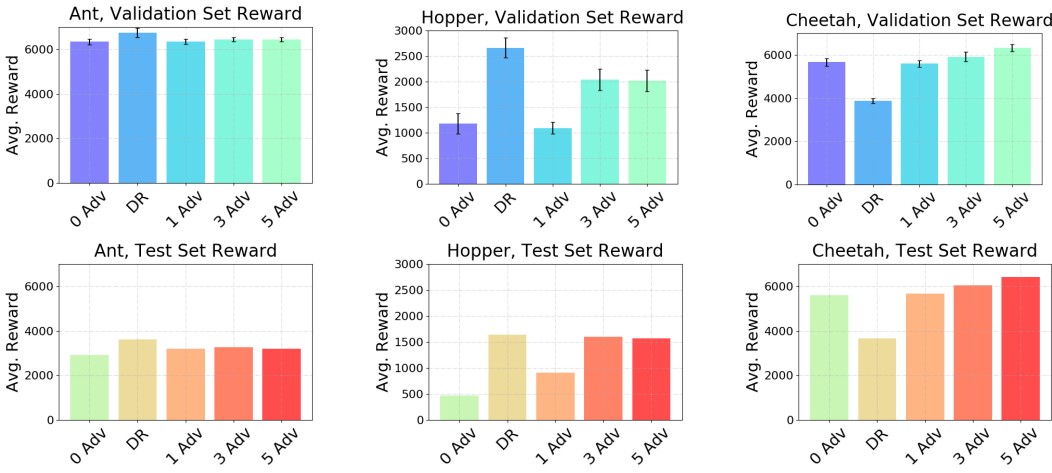

Figure 6: Average reward for Ant, Hopper, and Cheetah environments across ten seeds and across the validation set (top row) and across the holdout test set (bottom row). We compare vanilla PPO, the domain randomization oracle, and the minimax adversary against RAP of size three and five. Bars represent the mean and the arms represent the std. deviation. Both are computed over 20 rollouts for each test-set sample. The std. deviation for the test set are not reported here for visual clarity due to the large variation in holdout test difficulty.

## C  HOLDOUT TESTS

In this section we describe in detail all of the holdout tests used.

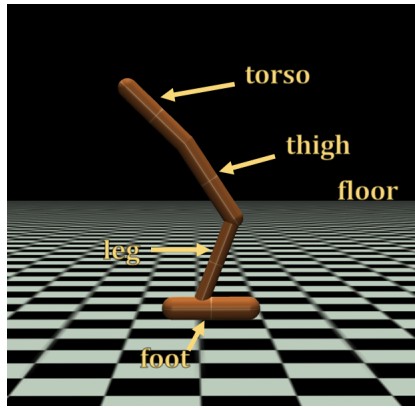

Figure 7: Labelled Body Segments of Hopper

Table 2: Hopper Holdout Test Descriptions

| Test | Body with Friction Coeff 1.3 | Body with Friction Coeff 0.7 |
|---|---|---|
| A | Torso, Leg | Floor, Thigh, Foot |
| B | Floor, Thigh | Torso, Leg, Foot |
| C | Foot, Leg | Floor, Torso, Thigh |
| D | Torso, Thigh, Floor | Foot, Leg |
| E | Torso, Foot | Floor, Thigh, Leg |
| F | Floor, Thigh, Leg | Torso, Foot |
| G | Floor, Foot | Torso, Thigh, Leg |
| H | Thigh, Leg | Floor, Torso, Foot |

## C.1 HOPPER

The Mujoco geom properties that we modified are attached to a particular body and determine its appearance and collision properties. For the Mujoco holdout transfer tests we pick a subset of the hopper 'geom' elements and scale the contact friction values by maximum friction coefficient, 1.3. Likewise, for the rest of the 'geom' elements, we scale the contact friction by the minimum value of 0.7. The body geoms and their names are visible in Fig. 7.

The exact combinations and the corresponding test name are indicated in Table 2 for Hopper.

## C.2 CHEETAH

The Mujoco geom properties that we modified are attached to a particular body and determine its appearance and collision properties. For the Mujoco holdout transfer tests we pick a subset of the

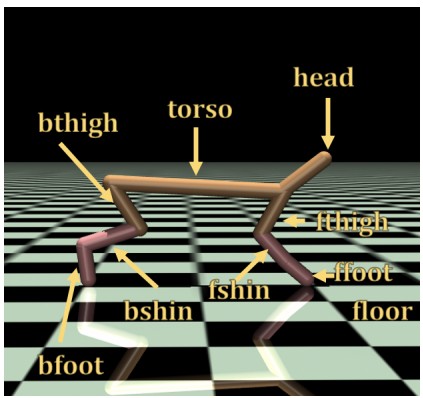

Figure 8: Labelled Body Segments of Cheetah

Table 3: Cheetah Holdout Test Descriptions. Joints in the table receive the maximum friction coefficient of 0.9. Joints not indicated have friction coefficient 0.1

| Test | Geom with Friction Coeff 0.9 |
|------|------------------------------|
| A | Torso, Head, Fthigh |
| B | Floor, Head, Fshin |
| C | Bthigh, Bshin, Bfoot |
| D | Floor, Torso, Head |
| E | Floor, Bshin, Ffoot |
| F | Bthigh, Bfoot, Ffoot |
| G | Bthigh, Fthigh, Fshin |
| H | Head, Fshin, Ffoot |

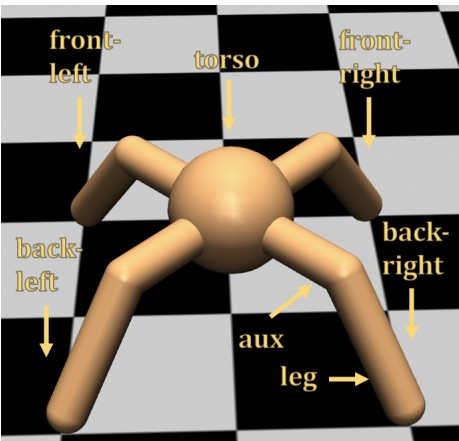

Figure 9: Labelled Body Segments of Ant

cheetah 'geom' elements and scale the contact friction values by maximum friction coefficient, 0.9. Likewise, for the rest of the 'geom' elements, we scale the contact friction by the minimum value of 0.1. The body geoms and their names are visible in Fig. 8.

The exact combinations and the corresponding test name are indicated in Table 4 for Hopper.

## C.3 ANT

We will use torso to indicate the head piece, leg to refer to one of the four legs that contact the ground, and 'aux' to indicate the geom that connects the leg to the torso. Since the ant is symmetric we adopt a convention that two of the legs are front-left and front-right and two legs are back-left and back-right. Fig. 9 depicts the convention. For the Mujoco holdout transfer tests we pick a subset of the ant 'geom' elements and scale the contact friction values by maximum friction coefficient, 0.9. Likewise, for the rest of the 'geom' elements, we scale the contact friction by the minimum value of 0.1.

Table 4: Ant Holdout Test Descriptions. Joints in the table receive the maximum friction coefficient of 0.9. Joints not indicated have friction coefficient 0.1

| Test | Geom with Friction Coeff 0.9 |
|------|------------------------------|
| A | Front-Leg-Left, Aux-Front-Left, Aux-Back-Left |
| B | Torso, Aux-Front-Left, Back-Leg-Right |
| C | Front-Leg-Right, Aux-Front-Right, Back-Leg-Left |
| D | Torso, Front-Leg-Left, Aux-Front-Left |
| E | Front-Leg-Left, Aux-Front-Right, Aux-Back-Right |
| F | Front-Leg-Right, Back-Leg-Left, Aux-Back-Right |
| G | Front-Leg-Left, Aux-Back-Left, Back-Leg-Right |
| H | Aux-Front-Left, Back-Leg-Right, Aux-Back-Right |

| Test Name | 0 Adv | 1 Adv | 3 Adv | Five Adv | Domain Rand |
|---|---|---|---|---|---|
| Test A | $410 \pm 140$ | $1170 \pm 570$ | $\mathbf{2210 \pm 630}$ | $2090 \pm 920$ | $1610 \pm 310$ |
| Test B | $430 \pm 150$ | $1160 \pm 540$ | $\mathbf{2240 \pm 730}$ | $2200 \pm 880$ | $1610 \pm 290$ |
| Test C | $560 \pm 120$ | $490 \pm 150$ | $610 \pm 250$ | $580 \pm 120$ | $\mathbf{1660 \pm 260}$ |
| Test D | $420 \pm 150$ | $1140 \pm 560$ | $\mathbf{2220 \pm 680}$ | $2130 \pm 890$ | $1612 \pm 360$ |
| Test E | $550 \pm 120$ | $500 \pm 150$ | $600 \pm 240$ | $590 \pm 120$ | $\mathbf{1680 \pm 280}$ |
| Test F | $420 \pm 150$ | $1200 \pm 620$ | $2080 \pm 750$ | $\mathbf{2160 \pm 890}$ | $1650 \pm 360$ |
| Test H | $560 \pm 130$ | $500 \pm 140$ | $600 \pm 230$ | $600 \pm 140$ | $\mathbf{1710 \pm 370}$ |
| Test G | $420 \pm 150$ | $1160 \pm 590$ | $\mathbf{2210 \pm 680}$ | $2160 \pm 920$ | $1560 \pm 340$ |

Table 5: Results on holdout tests for each of the tested approaches for Hopper. Bolded values have the highest mean

| Test Name | 0 Adv | 1 Adv | 3 Adv | Five Adv | Domain Rand |
|---|---|---|---|---|---|
| Test A | $4400 \pm 2160$ | $5110 \pm 730$ | $4960 \pm 1280$ | $\mathbf{5560 \pm 1060}$ | $2800 \pm 1540$ |
| Test B | $6020 \pm 880$ | $5980 \pm 290$ | $6440 \pm 1620$ | $\mathbf{6880 \pm 1090}$ | $3340 \pm 600$ |
| Test C | $5880 \pm 1030$ | $5730 \pm 640$ | $\mathbf{6740 \pm 1190}$ | $6410 \pm 790$ | $4280 \pm 240$ |
| Test D | $5990 \pm 940$ | $5960 \pm 260$ | $6430 \pm 1610$ | $\mathbf{6880 \pm 1090}$ | $3360 \pm 570$ |
| Test E | $5570 \pm 570$ | $5670 \pm 290$ | $5800 \pm 1316$ | $\mathbf{6530 \pm 1250}$ | $3720 \pm 540$ |
| Test F | $5870 \pm 750$ | $5800 \pm 350$ | $6500 \pm 1100$ | $\mathbf{6770 \pm 1070}$ | $3810 \pm 330$ |
| Test H | $5310 \pm 1060$ | $5270 \pm 700$ | $5610 \pm 720$ | $\mathbf{5660 \pm 980}$ | $4560 \pm 560$ |
| Test G | $5710 \pm 650$ | $5790 \pm 300$ | $5890 \pm 1240$ | $\mathbf{6560 \pm 1240}$ | $3380 \pm 720$ |

Table 6: Results on holdout tests for each of the tested approaches for Half Cheetah. Bolded values have the highest mean

The exact combinations and the corresponding test name are indicated in Table 4 for Hopper.

# D  RESULTS

Here we recompute the values of all the results and display them with appropriate standard deviations in tabular form.

There was not space for the ant validation set results so they are reproduced here.

| Test Name | 0 Adv | 1 Adv | 3 Adv | Five Adv | Domain Rand |
|---|---|---|---|---|---|
| Test A | $590 \pm 650$ | $730 \pm 630$ | $600 \pm 440$ | $560 \pm 580$ | $\mathbf{900 \pm 580}$ |
| Test B | $5240 \pm 280$ | $5530 \pm 200$ | $5770 \pm 100$ | $5710 \pm 180$ | $\mathbf{6150 \pm 180}$ |
| Test C | $750 \pm 820$ | $1090 \pm 660$ | $1160 \pm 540$ | $1040 \pm 760$ | $\mathbf{1370 \pm 800}$ |
| Test D | $5220 \pm 300$ | $5560 \pm 220$ | $5770 \pm 90$ | $5660 \pm 190$ | $\mathbf{6120 \pm 180}$ |
| Test E | $5270 \pm 290$ | $5570 \pm 210$ | $5770 \pm 100$ | $5660 \pm 220$ | $\mathbf{6140 \pm 150}$ |
| Test F | $780 \pm 860$ | $1160 \pm 570$ | $1120 \pm 580$ | $1140 \pm 870$ | $\mathbf{1390 \pm 750}$ |
| Test H | $130 \pm 290$ | $420 \pm 300$ | $210 \pm 220$ | $160 \pm 270$ | $\mathbf{700 \pm 560}$ |
| Test G | $5290 \pm 280$ | $5560 \pm 220$ | $5770 \pm 100$ | $5700 \pm 190$ | $\mathbf{6150 \pm 160}$ |

Table 7: Results on holdout tests for each of the tested approaches for Ant. Bolded values have the highest mean

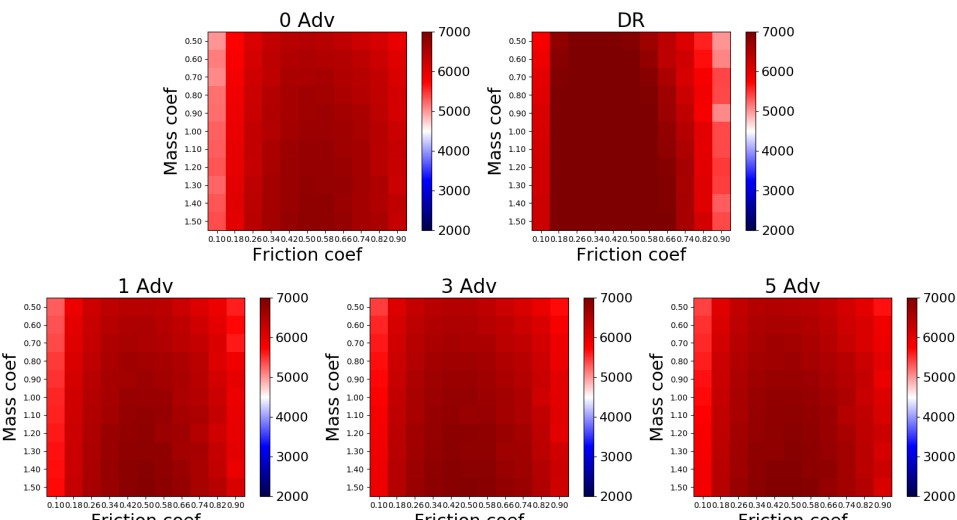

Figure 10: Ant Heatmap: Average reward across 10 seeds on each validation set (mass, friction) parametrization.

# E   CHALLENGES OF DOMAIN RANDOMIZATION

In our experiments, we find that naive parametrization of domain randomization can result in a brittle policy, even when evaluated on the same distribution it was trained on.

**Effect of Domain Randomization Parametrization**

From Fig. 6, we see that in the Ant and Hopper domains, the DR oracle achieves the highest transfer reward in the validation set as expected since the DR oracle is trained directly on the validation set. Interestingly, we found that the domain randomization policy performed much worse on the Half Cheetah environment, despite having access to the mass and friction coefficients during training. Looking at the performance for each mass and friction combination in Fig. 11, we found that the DR agent was able to perform much better at the low friction coefficients and learned to prioritize those values at the cost of significantly worse performance on average. This highlights a potential issue with domain randomization: while training across a wide variety of dynamics parameters can increase robustness, naive parametrizations can cause the policy to exploit subsets of the randomized domain and lead to a brittle policy. This is a problem inherent to the expectation across domains that is used in domain randomization; if some subset of randomizations have sufficiently high reward the agent will prioritize performance on those at the expense of robustness.

We hypothesize that this is due to the DR objective in Eq. 2 optimizing in expectation over the sampling range. To test this, we created a separate range of 'good' friction parameters $[0.5, 1.5]$ and compared the robustness of a DR policy trained with 'good' range against a DR policy trained with 'bad' range $[0.1, 0.9]$ in Fig. 11. Here we see that a 'good' parametrization leads to the expected result where domain randomization is the most robust. We observe that domain randomization underperforms adversarial training on the validation set despite the validation set literally constituting the training set for domain randomization. This suggests that underlying optimization difficulties caused by significant variations in reward scaling are partially to blame for the poor performance of domain randomization. Notably, the adversary-based methods are not susceptible to the same parametrization issues.

**Alternative DR policy architecture**
As discussed above and also identified in Rajeswaran et al. (2016), the expectation across randomizations that is used in domain randomization causes it to prioritize a policy that performs well in a high-reward subset of the randomization domains. This is harmless when domain randomization is used for randomizations of state, such as color, where all the randomization environments have the same expected reward, but has more pernicious effects in dynamics randomizations. Consider a set of $N$ randomization environments, $N - 1$ of which have reward $R_{\text{low}}$ and one of which has

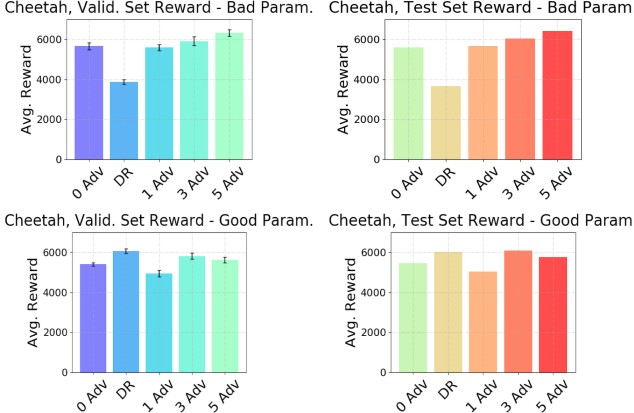

Figure 11: Average reward for Half Cheetah environment across ten seeds. Top row shows the average reward when trained with a 'bad' friction parametrization which lead to DR not learning a robust agent policy, and bottom row shows the average reward when trained with a 'good' friction parametrization.

has reward $R_{\text{high}}$ where $R_{\text{high}} >> R_{\text{low}}$. If the agent cannot identify which of the randomization environments it is in, the intuitively optimal solution is to pick the policy that optimizes the high reward environment. One possible way out of the quandary is to use an agent that has some memory, such as an LSTM-based policy, thus giving the possibility of identifying which environment the agent is in and deploying the appropriate response. However, if $R_{\text{high}}$ is sufficiently large and there is some reduction in reward associated with performing the system-identification necessary to identify the randomization, then the agent will not perform the system identification and will prioritize achieving $R_{\text{high}}$. As an illustration of this challenge, Fig. 12 compares the results of domain randomization on the half-cheetah environment with and without memory. In the memory case, we use a $64$ unit LSTM. As can be seen, there is an improvement in the ability of the domain randomized policy to perform well on the full range of low-friction / high mass values, but the improved performance does not extend to higher friction values. In fact, the performance contrast is enhanced even further as the policy does a good deal worse on the high friction values than the case without memory.

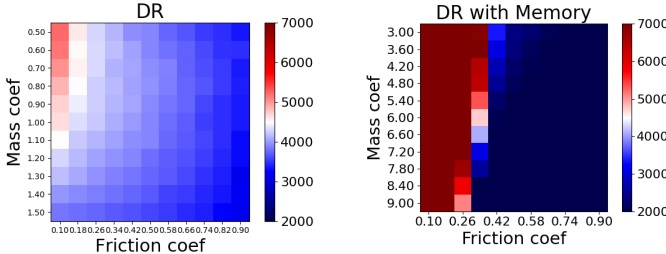

Figure 12: Left: heatmap of the performance of the half-cheetah domain randomized policy across the friction and mass value grid. Right: Left: heatmap of the performance of the half-cheetah domain randomized policy across the friction and mass value grid where the agent policy is an LSTM.

# F    ADDITIONAL EXPERIMENTS

Here we outline a few more experiments we ran that demonstrate the value of additional adversaries. We run the following tasks:

## F.1    DEEPMIND CONTROL CATCH

This task uses the same Markov Decision Process described in Sec. A. The challenge (Tassa et al., 2020), pictured in Fig. 13, is to get the ball to fall inside the cup. As in the other continuous control

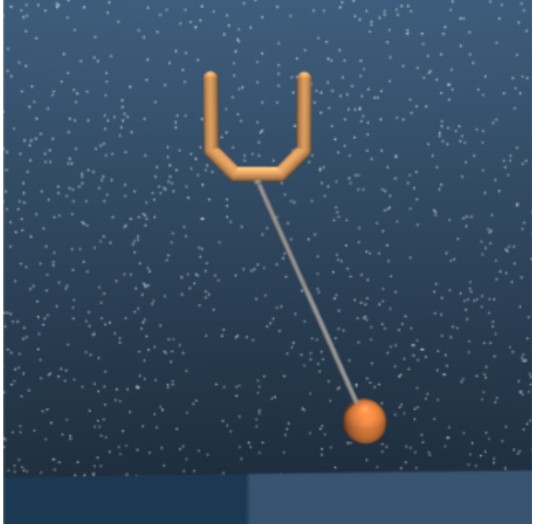

Figure 13: The DeepMind Control catch task. The cup moves around and attempts to get the ball to fall inside.

tasks, we apply the adversary to the actions of the agents (which is controlling the cup). We then test on variations of the mass of both the ball and the cup. The heatmaps for this task are presented in Fig. 14 where the 3 adversary case provides a slight improvement in the robustness region relative to the 1 adversary case.

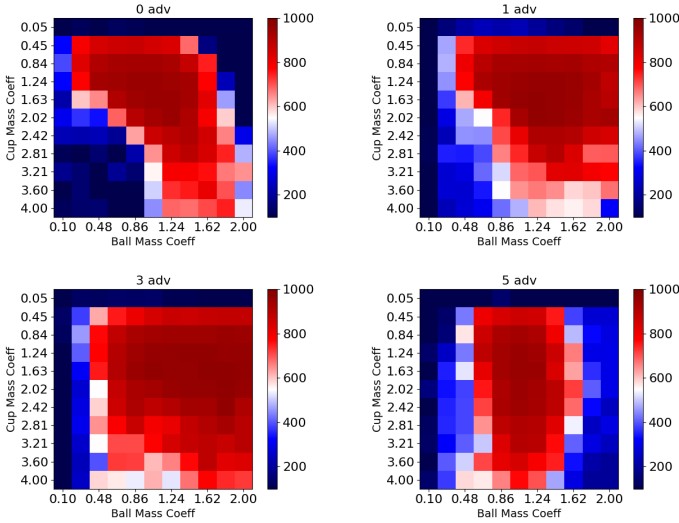

Figure 14: (Top left) 0 adversary, (top right) 1 adversary, (bottom left) 3 adversary, (bottom right) 5 adversaries for variations of cup and ball mass.

## F.2 MULTI-ARMED BERNOULLI BANDITS

As an illustrative example, we examine a multi-armed stochastic bandit, a problem widely studied in reinforcement learning literature. Generally, successful strategies for multi-arm bandit problems involve successfully balancing the exploration across arms and exploiting the 'best' arm. A "robust" strategy should have vanishing regret as the time horizon goes to infinity. We construct a 10-armed bandit where each arm $i$ is parametrized by a value $p$ where p is the probability of that arm returning

1. The goal of the agent is to minimize total cumulative regret $R_n$ over a horizon of $n$ steps:

$$R_n = n \max_i \mu_i - \mathbb{E}\left[\sum_{t=0}^{n} a_t\right]$$

where $a_t$ corresponds to picking a particular arm. At each step, the agent is given an observation buffer of stacked frames consisting of all previous (action, reward) pairs padded with zeros to keep the length fixed. The adversary has a horizon of 1; at time-step zero it receives an observation of 0 and outputs the probability for each arm. At the termination of the horizon the adversary receives the negative of the cumulative agent reward. For our domain randomization baseline we use uniform sampling of the $p$ value for each arm. We chose a horizon length of $T = 100$ steps. The MDP of the agent is characterized as follows:

- $s_t = \left[0^{n*(T-t)\times 1}, r_t, a_t, r_{t-1}, a_{t-1}, \ldots, r_0, a_0\right]$
- $r_t = X(a_i) - \max_i \mu_i$
- $a_t^{\text{agent}} \in 0 \ldots 9$

At each step, the agent is given an observation buffer of stacked frames consisting of all previous (action, reward) pairs. The buffer matching the horizon length is padded with zeros. For each training step, the agent receives a reward of the negative expected regret. We set up the adversary problem as an MDP with a horizon of 1.

- $s_t = [0.0]$
- $r = -\sum_{i=1}^{T} r_t$
- $a^{\text{adv}} \in [0, 1]^{10}$

During adversarial training, we sample a random adversary at the beginning of each rollout, and allow it to pick 10 $p$ values that are then shuffled randomly and then assigned to each arm (this is to prevent the agent from deterministically knowing which arm has which $p$ value). The adversary is always given an observation of a vector of zeros and is rewarded once at the end of the rollout. We also construct a hold-out test of two bandit examples which we colloquially refer to as "evenly spread" and "one good arm." In "evenly spread", the arms, going from 1 to 10 have evenly spaced probabilities in steps of $0.1$ $0.1, 0.2, 0.3, \ldots 0.8, 0.9$. In "one good arm" 9 arms have probability $0.1$ and one arm has probability $0.9$. As our policy for the agent, we use a Gated Recurrent Unit network with hidden size 256.

An interesting feature of the bandit task is that it makes clear that the single adversary approach corresponds to training on a single, adversarially constructed bandit instance. Surprisingly, as indicated in Fig. 15, this does not perform terribly on our two holdout tasks. However, there is a clear improvement on both tasks in the four adversary case. All adversarial approaches outperform an Upper Confidence Bound-based expert (shown in red). Interestingly, domain randomization, which had superficially good reward at training time, completely fails on the "one good arm" holdout task. This suggests another possible failure mode of domain randomization where in high dimensions uniform sampling may just fail to yield interesting training tasks. Finally, we note that since the upper confidence approach only tries to minimize regret asymptotically, our outperforming it may simply be due to our relatively short horizon; we simply provide it as a baseline.

## G  COST AND HYPERPARAMETERS

Here we reproduce the hyperparameters we used in each experiment and compute the expected runtime and cost of each experiment. Numbers indicated in {} were each used for one run. Otherwise the parameter was kept fixed at the indicated value.

### G.1  HYPERPARAMETERS

For Mujoco the hyperparameters are:

- Learning rate:

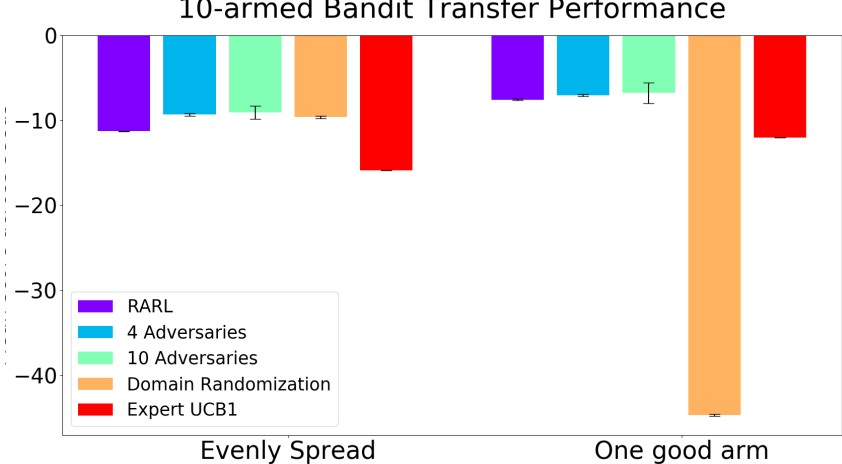

Figure 15: Two transfer tests for the bandit task. On both tasks the 4 adversary case has improved performance relative to RARL while domain randomization performs terribly on all tasks. Bars indicate one std. deviation of the performance over 100 trials.

- $\{.0003, .0005\}$ for half cheetah
- $\{.0005, .00005\}$ for hopper and ant
- Generalized Advantage Estimation $\lambda$
  - $\{0.9, 0.95, 1.0\}$ for half cheetah
  - $\{0.5, 0.9, 1.0\}$ for hopper and ant
- Discount factor $\gamma = 0.995$
- Training batch size: 100000
- SGD minibatch size: 640
- Number of SGD steps per iteration: 10
- Number of iterations: 700
- We set the seed to 0 for all hyperparameter runs.
- The maximum horizon is 1000 steps.

For the validation across seeds we used 10 seeds ranging from 0 to 9. All other hyperparameters are the default values in RLlib Liang et al. (2017) 0.8.0

## G.2 COST

For all of our experiments we used AWS EC2 c4.8xlarge instances which come with 36 virtual CPUs. For the Mujoco experiments, we use 2 nodes and 11 CPUs per hyper-parameter, leading to one full hyper-parameter sweep fitting onto the 72 CPUs. We run the following set of experiments and ablations, each of which takes 8 hours.

- 0 adversaries
- 1 adversary
- 3 adversaries
- 5 adversaries
- Domain randomization

for a total of 5 experiments for each of Hopper, Cheetah, Ant. For the best hyperparameters and each experiment listed above we run a seed search with 6 CPUs used per-seed, a process which takes about 12 hours. This leads to a total of $2*8*5*3+2*12*3*5 = 600$ node hours and $36*600 \approx 22000$

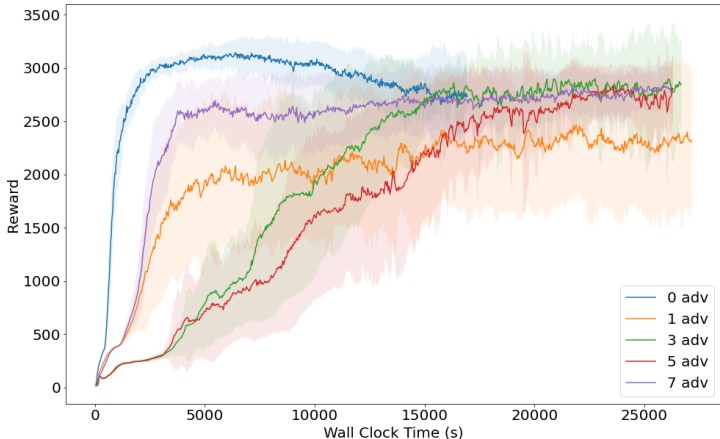

Figure 16: Wall-clock time vs. reward for varying numbers of adversaries. Despite varying adversary numbers, the wall-clock time of 1, 3, 5, and 7 adversary runs are all the same.

CPU hours. At a cost of $\approx 0.3$ dollars per node per hour for EC2 spot instances, this gives $\approx 180$ dollars to fully reproduce our results for this experiment. If the chosen hyperparameters are used and only the seeds are sweep, this is $\approx 100$ dollars.

### G.3 RUN TIME AND SAMPLE COMPLEXITY

Here we briefly analyze the expected run-time of our algorithms. While there is an additional cost for adding a single adversary equal to the sum of the cost of computing gradients at train time and actions at run-time for an additional agent, there is no additional cost for adding additional adversaries. Since we divide the total set of samples per iteration amongst the adversaries, we compute approximately the same number of gradients and actions in the many-adversary case as we do in the single adversary case. In Fig. 16 plot of reward vs. wall-clock time supports this argument: the 0 adversary case runs the fastest but all the different adversary numbers complete 700 iterations of training in approximately the same amount of time. Additionally, Fig. 17 demonstrates that there is some variation in sample complexity but the trend is not consistent across adversary number.

### G.4 CODE

Our code is available at **ANONYMIZED**. For our reinforcement learning code-base we used RLlib Liang et al. (2017) version 0.8.0 and did not make any custom modifications to the library.

## H PURE NASH EQUILIBRIA DO NOT NECESSARILY EXIST

While there are canonical examples of games in which pure Nash equilibria do not exist such as rock-paper-scissors, we are not aware one for sequential games with continuous actions. Tessler et al. (2019) contains an example of a simple, horizon 1 MDP where duality is not satisfied. The pure minimax solution does not equal the value of the pure maximin solution and a greater value can be achieved by randomizing one of the policies showing that there is no pure equilibrium.

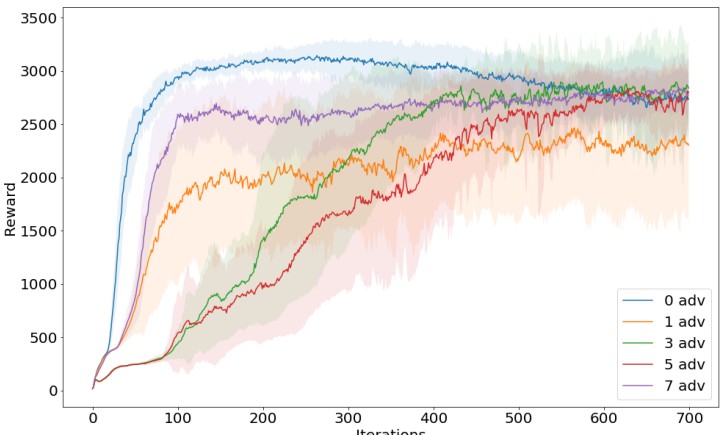

Figure 17: Iterations vs. reward for varying numbers of adversaries. Despite varying adversary numbers, the wall-clock time of 1, 3, 5, and 7 adversary runs are all the same.

