# OpenReview forum: "Robust Reinforcement Learning using Adversarial Populations"
_ICLR.cc/2021/Conference — Reject_

### Official Review · AnonReviewer1 · 2020-10-26
**Simple and nice idea, but experiments are not convincing enough.**

**Rating:** 5
**Confidence:** 4

**Review:**

Summary: This paper proposes to improve robustness in reinforcement learning via a population of diverse adversaries, where previous works mainly focus on the use a single adversary to mitigate the problem that the trained policy could be highly exploitable by the adversary. Specifically, at each iteration, it randomly selects an adversary from the population for rollouts, and it is trained by PPO. Experiments are conducted on 3 MuJoCo environments in comparison with vanilla PPO, domain randomization.

Strong points: Using a population of adversaries to improve robustness in RL is interesting. The idea is simple, and the writing is clear.

Concerns:
My major concern is in the experimental evaluation.
a. Results are shown using final performance. I am curious about the learning curve – how does the method compare against other baselines in terms of sample efficiency? A side-effect using a population is that RAP needs to update n adversaries at each training iteration compared with using a single adversary, and will incur more computation overhead. Could authors fairly compare with other baselines in terms of this and show the learning curve?

b. How *MUCH LONGER* does it take to run RAP compared with other baselines? How much more memory does it take to use n adversaries compared with a single adversary?

c. Could authors compare with a naive extension of the single adversary case in which the single adversary sample n actions? Is the baseline comparable with RAP using n adversaries?

d. I am confused why RAP is built upon an on-policy algorithm. A number of works using population-based methods are built upon off-policy algorithms as agents in the population can share the samples and could be beneficial. Could authors build the method upon off-policy algorithms to further improve the applicability of RAP?

e. For Figure 3, the performance gain over using a single adversary is not significant on HalfCheetah and Ant, and the results is not convincing enough to support the claim.

As the paper uses the population-based methods, it is also worth discussing its relation with Khadka et al. 2018, etc.

---

> ### Author Response · Authors · 2020-11-16
> **Additional experiments added, graphs on memory and reward curves added**
>
> We thank the reviewer for their detailed review and useful feedback and are happy the reviewer finds our method interesting. We believe that the following clarifications and answers to the reviewer’s questions that were raised here will make for a stronger paper. We have addressed the individual questions below and made changes to the paper (marked in red) to help address these issues. In particular, we failed to properly address sample complexity and run-time due to the page limit but are using the extra space to address this.
> >  a. Results are shown using final performance. I am curious about the learning curve – how does the method compare against other baselines in terms of sample efficiency? A side-effect using a population is that RAP needs to update n adversaries at each training iteration compared with using a single adversary, and will incur more computation overhead. Could authors fairly compare with other baselines in terms of this and show the learning curve?
>
> A: This is a fair point; we initially addressed this in the paper but were limited on space. Fortunately, the usage of RAP incurs no additional run-time cost relative to using a single adversary although it does incur a small memory cost in storing each of the additional adversary policies. As we believe the closest baseline is the original RARL paper, we have added Figure 11 to the appendix to compare the learning curve across number of adversaries (for hopper, cheetah, and ant). While there is some random variation in how quickly the agent learns (likely due to small grid searches over hyperparameters), the overall training speed seems to vary only a little in terms of number of iterations.
> Importantly, we also want to point out that the maximum reward achieved in the 0 adversary is higher than any of the adversarial cases. This is the behavior that should be expected given an adversary as the cases playing against an adversary should have lower converged reward than the maximum possible reward without an adversary.
> > b. How MUCH LONGER does it take to run RAP compared with other baselines? How much more memory does it take to use n adversaries compared with a single adversary?
>
> A: Because we divide the number of samples approximately evenly between the adversaries and mini-batching is generally used in RL, the actual run-time of our training is actually equivalent to that of training a single adversary. We have added Fig. 10 and Fig. 11 to the appendix which compares the reward curves both as a function of number of iterations and on wall-clock time. From the wall-clock figure, it should be clear that there is no additional slow-down from using additional adversaries once you are already using 1 adversary. For N adversaries, the additional memory cost is the cost of storing the N-1 extra neural networks. Since RL neural network policies are small in continuous control (generally 2-3 layers of hidden size that rarely exceeds 400), this is on the order of megabytes.
> > c. Could authors compare with a naive extension of the single adversary case in which the single adversary sample n actions? Is the baseline comparable with RAP using n adversaries?
>
> A: We think this is a really good variant of the experiment as it is an alternative way to represent the mixed nash equilibrium. However, it may be difficult to get the agent to actually take advantage of the randomization. In multi-agent systems, agents have a tendency to cycle through pure strategies even if the Nash equilibrium is mixed (as established in papers like “Learning with opponent-learning awareness” by Foerster et al). While we have not run this particular experiment, it is suggested in our future work section.
> > d. I am confused why RAP is built upon an on-policy algorithm. A number of works using population-based methods are built upon off-policy algorithms as agents in the population can share the samples and could be beneficial. Could authors build the method upon off-policy algorithms to further improve the applicability of RAP?
>
> A: This is a good idea as we agree that sharing samples could be beneficial, although it is also possible that sharing samples could reduce the diversity of our adversary pool! We used on-policy methods because it’s empirically (in our experience) more stable for multi-agent problems and we do not have to use explicit, special multi-agent methods such as MADDPG to handle non-stationarity issues. Additionally, the original Robust Adversarial Reinforcement Learning paper uses TRPO, so we chose another on-policy method for a closer comparison.

---

> > ### Author Response · Authors · 2020-11-16
> > **Rebuttal Part 2**
> >
> > > e. For Figure 3, the performance gain over using a single adversary is not significant on HalfCheetah and Ant, and the results is not convincing enough to support the claim.
> >
> > A. We have added an additional non-walking experiment in continuous control environments and a bandit environment that demonstrate the value of this approach in the Appendix. We believe the challenge is that several of these mujoco environments, in particular ant, are inherently quite stable and are not particularly affected by the perturbations we apply whereas hopper is a bit more unstable due to the single contact point. Thus, there is little performance improvement in robustness when adversaries are added in the first place and not much room for improvement from additional adversaries in the two non-hopper experiments. To address this point directly, we have also added a ball-in-cup task  and as an example that we think makes the failure modes of a single adversary clearer, we have added a 10 dimensional Bernoulli bandit task where each adversary outputs a bandit instance that the agent is allowed to play in. Here the use of a single adversary corresponds to a single bandit instance and the issue of overfitting is clear in that the agent will memorize the single bandit instance whereas the generalization to new instances is better when multiple adversaries are used. This example also makes clear the challenge of domain randomization in high dimensions, namely that most samples are uninformative when sampled uniformly. Consequently, the domain randomization trained policy completely fails to sample arms appropriately on one of our test bandit instances though it looks superficially good on uniformly sampled instances; in contrast the agent trained against adversarially selected bandit instances is robust on our tests.
> >
> > >As the paper uses the population-based methods, it is also worth discussing its relation with Khadka et al. 2018, etc.
> >
> > A: We are not sure which paper the reviewer is referencing but we would love to read it and appropriately reference it. Would you mind providing the full reference?
> >
> > Finally, we have slightly re-written the experiments section of the paper to clarify which tasks are in-distribution vs. out-of-distribution. Playing against tasks generated by our adversary class are in-distribution, whereas playing against mass and friction variations is out-of-distribution. Playing an adversary only provides guarantees on in-distribution performance and Figure 2 now shows that agents trained against a single adversary do not inherit this in-distribution performance guarantee for the reasons identified in the introduction. Additionally, we now test how domain randomization does on these in-distribution tasks and show that domain randomization does not provide robustness on these tasks either.
> > As the discussion of failure modes of domain randomization is interesting, but not the key contribution of the paper, we have moved the section formerly called "Hypothesis 3" into the appendix and direct readers there in the experiments section.
> > To summarize, we have made the following changes and additions (highlighted in red in the latest paper version):
> > - Compared the performance of agents across different adversary training regimes (agent trained against 1 adv vs. adversaries from the 3 adversary case, domain randomization agents against 1 and 3 adversaries, etc.). This is summarized in Fig. 2.
> > - We have added additional experiments on bandit tasks and a Deepmind control ball-in-cup task to demonstrate our method in more varied domains.
> > - We have rewritten the experiment section to indicate which tasks are in-distribution (for which we have lower bounded performance guarantees from adversarial training) vs. out-of-distribution (for which performance improvements are plausible but not guaranteed).
> > - We have moved the discussion on failure modes of domain randomization to the appendix and added an additional example of a failure mode of domain randomization.
> > - Edits to respond to particular writing issues / notation brought up by the reviewers.
> >
> > We hope that these changes address the main concerns brought up by the reviewers and are happy to respond to any further questions.

---

### Official Review · AnonReviewer4 · 2020-10-28
**A clean formulation with some compelling results**

**Rating:** 7
**Confidence:** 4

**Review:**

#### Summary

The authors present a scheme that can be used to train agents to be robust against a population of adversarial policies, in which adversaries can perturb actions via an additive perturbation.  Motivated by the observation that agents trained against a single policy may overfit to that policy and hence will lack robustness to new/unseen policies, the authors seek to show that their method generalizes well to unseen policies at test time.  Their experiments consider several simulated environments, in which they show generally good performance against several baselines.

#### Strengths

- I find the argument that agents will overfit to a single policy convincing.  While the motivating example WRT different forces acting on an agent may not be a scenario that robustness against adaptive perturbations can handle, the general claim seems to hold water.  That is, it seems plausible that an agent might receive high reward in a zero sum min-max game subject to a single adversary simply because the adversary is not strong enough to limit the cumulative reward that agent receives.

- The formulation for RAP is presented very clearly.  It is quite helpful to first consider the single minmax adversary and domain randomization formulations, both of which seemed to have played roles in the development of RAP.  Indeed, this seems a very natural way of formulating the problem.  More generally, the paper is quite well written.

- The experiments, and in particular Fig 2, clearly demonstrate the utility of this approach.  One can see a notable difference when an agent is trained with respect to three adversarial policies vs. when it is only trained with a single adversarial policy.

#### Weaknesses

- The utility of this approach is less clear when one considers Fig. 3.  It seems that domain randomization often outperforms RAP.  In my opinion, this weakens part of the claim made by the authors.  However, it does seem true that given that DR is designed to perform well on new domains _in expectation_, it may be preferable to use RAP when one suffers worst case dynamics changes (e.g. Fig. 5).

- Further study should be done as to how many adversaries one needs to provide meaningful levels of robustness.  It seems that in different experiments, 1, 3, and 5 adversaries were considered.  How should one decided how many adversaries to use?

- I felt the paper was a bit unclear about what states are available to train adversarial and regular policies.  That is, it is unclear whether in the trajectories $\tau_j$ and $\tau_j^i$, the agents/adversaries observe their own actions, the actions of their counterpart (e.g. the agent observing the adversaries action), or both agent and adversary observing the perturbed action $a + \alpha \bar{a}$.  The latter case would lead to a lack of observability for both agent and adversary.  Perhaps the authors can clarify this in the rebuttal.

- One weakness is that only one scenario (e.g. walking within these three environments) was considered.  It seems that the claims of the paper could be strengthened if more environments/tasks were considered.

#### Further questions/clarifications

- The reward function seems to be denoted as $\mathcal{R}$, $R$, and $r$ in various places.

- The bolding in Table 1 is a bit confusing.  I would be fairer to bold the most robust approach overall, rather than the most robust approach of the methods you propose.

- In the notation of Section 4, why are the rollouts of length $M$ rather than $T$, as indicated in the formulation?

#### Final Thoughts

Overall, I thought this was a solid and interesting paper.  The motivation is compelling, the formulation is relatively clean, and the experiments generally back up the claims that are made by the authors.  There are a few weaknesses, as I enumerated above, but even still I think that this is a valuable contribution.

---

> ### Author Response · Authors · 2020-11-16
> **Notation mistakes corrected, additional experiments added, MDP formulation added, review appreciated!**
>
> We thank the reviewer for their detailed and thoughtful review, and are happy that the reviewer finds our contribution compelling and interesting. We address some main concerns as follows:
>
> > “I felt the paper was a bit unclear about what states are available to train adversarial and regular policies. That is, it is unclear whether in the trajectories and , the agents/adversaries observe their own actions, the actions of their counterpart (e.g. the agent observing the adversaries action), or both agent and adversary observing the perturbed action ”
>
> Due to limited space, we had moved the formulation of the MDP into Appendix A. To clarify, the agent sees its observation and its own prior action which gives it some amount of adaptability to the mismatched dynamics. The adversary only sees the agent state as we want the adversaries to represent a mismatched dynamical system. Allowing the adversary to also see the agent action would be similar to conditioning the adversary directly on the agent policy, which is not often a characteristic of physical dynamical systems.
>
> >Further study should be done as to how many adversaries one needs to provide meaningful levels of robustness. It seems that in different experiments, 1, 3, and 5 adversaries were considered. How should one decided how many adversaries to use?
>
> Due to a lack of space, we are not able to put all the graphs that support this point into the main body of the text. However, we find empirically that three adversaries is usually optimal for the domains we consider and show a sweep from 0 - 11 adversaries for the hopper domain in the Appendix B . We find that once the adversary number increases significantly, the generalization performance starts to degrade. We hypothesize that this is due to the decreasing total number of samples per iteration allotted to each adversary. We note, however, that since the goal of the adversaries is to cover possible failure modes and approximate mixed nash, more than three adversaries may be needed for problems with many failure modes.
>
> > “One weakness is that only one scenario (e.g. walking within these three environments) was considered. It seems that the claims of the paper could be strengthened if more environments/tasks were considered.”
>
> We have added an additional set of experiments to the Appendix that we think supports our point that a single adversary is often insufficient. We have added a robotic arm task involving catching a ball in a cup and a Bernoulli bandit experiment in which the adversary takes in a zero vector and outputs the Bernoulli parameter p for each of ten arms. In particular, we would like to highlight that the bandit experiment makes very clear some of the weaknesses of domain randomization, as shown in Figure 15. Namely, that DR performs superficially well on uniformly sampled tasks (here evenly spread corresponds to just sampling the value p from the uniform distribution which is exactly the sampling distribution we use for domain randomization) but can fail catastrophically on hard tasks that are not often surfaced by uniform sampling (here “one good arm” which corresponds to one arm having high bernoulli value p and the others all having very low value).
>
> **Notation**
> We appreciate the reviewer for catching the mistakes in reward function notation and rollout length. We have corrected these in the new uploaded version (in red) and appreciate the careful readthrough.
> We chose to bold only the proposed method as domain randomization is technically outside the class of perturbations that our adversaries can construct and is consequently not a fair baseline. We provide domain randomization as an unfair baseline (essentially more of an oracle) to explicitly demonstrate limitations of our approach and show the possibility of improved performance if adversaries were constructed that somehow included the perturbations generated by domain randomization in their class of perturbations. We also wanted to include domain randomization as a way of pointing out limitations in this work, namely, that for the studied domains there are obvious domain randomizations that perform well (albeit with some downsides discussed in the paper). However, we understand that the presentation is unclear and have amended the table to show both best result overall and best within the adversarial methods.

---

> > ### Author Response · Authors · 2020-11-22
> > **Summary of changes**
> >
> > Finally, we have slightly re-written the experiments section of the paper to clarify which tasks are in-distribution vs. out-of-distribution. Playing against tasks generated by our adversary class are in-distribution, whereas playing against mass and friction variations is out-of-distribution. Playing an adversary only provides guarantees on in-distribution performance and Figure 2 now shows that agents trained against a single adversary do not inherit this in-distribution performance guarantee for the reasons identified in the introduction. Additionally, we now test how domain randomization does on these in-distribution tasks and show that domain randomization does not provide robustness on these tasks either.
> > As the discussion of failure modes of domain randomization is interesting, but not the key contribution of the paper, we have moved the section formerly called "Hypothesis 3" into the appendix and direct readers there in the experiments section.
> > To summarize, we have made the following changes and additions (highlighted in red in the latest paper version):
> > - Compared the performance of agents across different adversary training regimes (agent trained against 1 adv vs. adversaries from the 3 adversary case, domain randomization agents against 1 and 3 adversaries, etc.). This is summarized in Fig. 2.
> > - We have added additional experiments on bandit tasks and a Deepmind control ball-in-cup task to demonstrate our method in more varied domains.
> > - We have rewritten the experiment section to indicate which tasks are in-distribution (for which we have lower bounded performance guarantees from adversarial training) vs. out-of-distribution (for which performance improvements are plausible but not guaranteed).
> > - We have moved the discussion on failure modes of domain randomization to the appendix and added an additional example of a failure mode of domain randomization.
> > - Edits to respond to particular writing issues / notation brought up by the reviewers.
> >
> > We hope that these changes address the main concerns brought up by the reviewers and are happy to respond to any further questions.

---

> > ### Comment · AnonReviewer4 · 2020-11-23
> > **Rebuttal response: new experiments are solid, improve the argument of the paper**
> >
> > > Re: what states are available to train adversarial and regular policies
> >
> > Thanks for clarifying.
> >
> > > "We find that once the adversary number increases significantly, the generalization performance starts to degrade."
> >
> > While not unexpected, this is a good result and what one would hope to observe.  The paper is certainly made stronger by including this sort of discussion.
> >
> > > "We have added an additional set of experiments to the Appendix that we think supports our point that a single adversary is often insufficient."
> >
> > The cup and bandit experiments are solid, and reinforce the experiments already presented in the main text.  Further details concerning the cup experiment would be good for a future version, as the short discussion made it challenging to understand the task at hand.
> >
> > > Overall response:
> >
> > I do agree with some of the reviewers that the formulation here is in some sense incremental.  However, I still feel that the experiments are solid, and that the new experiments show that this algorithm is quite effective, especially when DR fails.  For this reason, I will stick to my original score.

---

### Official Review · AnonReviewer2 · 2020-10-30
**The proposed algorithm is good but the motivation is unsolid and experiments are limited**

**Rating:** 4
**Confidence:** 4

**Review:**

This paper proposes an algorithm to improve the robustness of reinforcement learning. The algorithm , RAP, combines ideas from domain randomization and adversarial training. Specifically, during learning, it trains an ensemble of adversary to attack the learner, with the hope that the learner can be robust to various situations. The experimental results show the proposed algorithm indeed outperform the respective baselines here (single-adversary training and domain randomization) in its ability to generalize the other test domains.

I think overall the proposed algorithm presents a simple and nice way to join the strengths of the adversary training and domain randomization. Indeed, we can view single-adversary training and domain randomization as special cases of RAP, which either uses a single adversary or not perform any update on the adversaries. For this, I would imagine RAP can be quite effective in practice.

* Issue about motivation and explaination

However, despite this nice design, I think the main motivation and the explanation why the proposed algorithm works are not completely correct. In introduction, the authors motivate the use of multiple adversary from that pure Nash equilibrium does not always exist in zero sum two player games. However, adversary training is not necessarily about solving the Nash equilibrium (which aims to find solutions such that minimax = maximin) but rather solving a maximin "only", whose solution is always well defined.

The motivating robot example is also quite misleading. In that case, the failure is due to the learner policy is not even solving maximin problem. Note in maximin, the adversary is chosen after knowing the learner's policy. This robot example is rather saying that minimax, where the learner optimizes after the adversary is chosen, is insufficient to generate robust behavior. However, this is not what adversarial training is about.

I think, it's probably because of this misunderstanding, the authors motivate the issue of the single-adversary training as the agent would overfit to the single adversary. Again this is due to incorrectly interpreting maximin in (1) as minimax.

Nonetheless, I do believe the proposed algorithm is effective in practice, but for a different reason from what the authors explain. I think the RAP does improve upon single-adversary training. Because it uses multiple randomly initialized adversarial policies, it may have a higher chance to overcome the non-convexity issue in the min part of maxmin and  therefore has a higher chance to find the maximin solution. In other words, the failure of the single-adversary implementation is most due to optimization difficulty not that the solution concept is incorrect. And the proposed algorithm is more of an optimization heuristic to better approximate the maximin problem (which actually is quite commonly used in the optimization literature). In fact, when given the learner policy, I believe by further taking the min among the multiple adversaries here and uses that for the learner update (i.e. the leaner would not use trajectories from all but the worst one), the robustness of the algorithm might further improve.

* Experiments

Given this work is lacking theoretical insights, I expect more experiments to be done to verify the proposed algorithm. The current paper only test RAP on mujoco environments. I think for understanding how well this algorithm works, the authors should test the algorithm on a more tasks with diverse characteristics (e.g. tabular, video games, tabular, traffic,) rather just the continuous robotics control domain.

In figure 2, I think a fairer comparison should let agents of both sides face the same adversary. Nonetheless, I agree that the performance plots later on are sufficient to show that RAP is better.

Lastly I like the in-depth discussion about the failure of domain randomization in halfcheetah, as it's also my main question when reading the previous part of the paper. However, I'm wondering if the bad performance is due to that the learner's policy is not expressive enough. I cannot find the description of the exact architecture used in the paper, but can you try training with a more expressive policy and see if domain randomization still perform worse than the direct training without the adversary?

---

> ### Author Response · Authors · 2020-11-16
> **Additional experiments, clarification about training objective**
>
> We thank the reviewer for their thoughtful feedback and constructive comments and are happy that the reviewer finds our proposed method to be effective and has a nice design. We hope to address the main concerns as follows:
> **Motivation and Explanation**
> We suspect that our use of notation in equation 1 and 3 has led to some lack of clarity on what training approach we are using to construct robust agents. In particular, our intent is to use multi-agent training to solve for the Nash equilibrium but the notation makes it appear as though we are trying to condition the agent’s policy on the antagonist to solve for a maximin solution. To be clear, the agent and adversary are being trained simultaneously, and neither conditions on either the actions or the policy of the other. We have updated the notation in Eq. (1) and (3) in a way that hopefully makes this clearer. Furthermore, we would like to outline more clearly why we think Nash is a reasonable solution concept here.
>  First, in the original Robust Adversarial Reinforcement Learning paper, the authors train their agent by alternating updates i.e. one gradient step for the agent, one gradient step for the adversary and repeating that process over and over. Since they are taking a single small gradient step rather than many, their approach should approach a Nash equilibrium. As this is the paper we are comparing against and trying to point out weaknesses in, it makes sense to use the same solution concept.
> Second, this is a zero sum game and consequently Nash, maximin, and minimax have the same value. While maximin would require solving a bi-level optimization problem, we can simply solve for Nash and have a guarantee of achieving the same value without the difficulty of bi-level optimization while providing the same robustness guarantees. Additionally, maximin can be viewed as an adversary conditioned on the agent policy. However, we are not looking for perturbations that are conditioned on the agent policy as we are simply trying to generate a perturbed set of dynamical systems. Since, for the continuous control tasks we study, the dynamics mismatch will not be conditioned on the agent policy, we do not want to learn adversaries that are aware of the agent policy.
>
> > “I think for understanding how well this algorithm works, the authors should test the algorithm on a more tasks with diverse characteristics (e.g. tabular, video games, tabular, traffic)”
>
> As suggested by the reviewers, we ran the following additional experiments and the results can be found in the appendix:
> - Added experiments in a different domain (multi-armed bandit)
> - Extended control experiments to non-walking tasks (catching a ball in cup)
>
> We chose to focus on continuous control experiments as the robustness challenges in this domain are primarily due to dynamics robustness. In many of the discrete domains, the robustness challenges often have to do with a lack of robustness to states that are outside the training distribution. For example, in Atari, it is well known that agent performance can be reduced by changing the background image (for example, see [1]). While robustness to state distribution mismatch is also an extremely interesting problem, it requires constructing a different class of adversaries that can apply perturbations to the state space. This is an open challenge and one we leave for future work. However, we hope that the additional continuous control environment in a distinct task and the bandit environments demonstrate that there is room for improvement over the single adversary case.
> In particular, the Bernoulli bandit example, in which adversaries output the p values (the probability of outputting 1), highlights the conceptual issue with using a single adversary. Here a single adversary corresponds to a single bandit instance and it seems sensible to expect that an optimal bandit sampling strategy will be learnt from interacting with just a single bandit. As an interesting side-note, this example also illustrates another case where domain randomization fails terribly. The test example “one good arm” (see Figure 15) corresponds to one arm with probability 0.9 and 9 arms with Bernoulli probability 0.1. Such an example, in which it is necessary to continue sampling all arms with non-zero probability to have a chance of identifying the good arm, appears rarely in the domain randomization training distribution. Instead, the domain randomization training distribution primarily features easy examples where most arms are reasonably good. Consequently, the domain randomization fails terribly on this adversarial example whereas agents trained against adversarially constructed bandit instances are robust.

---

> > ### Author Response · Authors · 2020-11-16
> > **Rebuttal Part 2**
> >
> > >”In figure 2, I think a fairer comparison should let agents of both sides face the same adversary”
> >
> > As the reviewer suggested, we also ran a more fair comparison of testing agents against the same adversary. We have uploaded a new version of Figure 2 that also includes the agent trained against 1 adversary playing the set of 3 adversaries and vice versa. The results show that the agent trained against 1 adversary performs terribly against the set of 3 adversaries while the reverse is not true. This suggests that the improved performance on the right hand side of Fig 2 (top right) is not due to weaker adversaries but due to a more robust agent.
> >
> > >”I cannot find the description of the exact architecture used in the paper, but can you try training with a more expressive policy and see if domain randomization still perform worse than the direct training without the adversary?”
> >
> > To clarify, all our agent and adversary policies consists of two layers with 64 units with a Tanh non-linearity. We assume that by a more expressive policy the reviewer meant a half-cheetah example in which the agent trained with DR has memory and could potentially identify which (friction, mass) pair it is operating in and subsequently perform optimally in this instance. We have provided additional experiments in Appendix E and argue that a more expressive policy does not alleviate a weakness of domain randomization being an expectation over the uniformly sampled (friction, mass) pairs. As an illustrative example, consider if one of the (friction, mass) pairs yielded an arbitrarily high score (eg. 1e20), and there was some minor cost associated with taking a few steps to identify which instance you were in, the agent would choose to not identify the (friction, mass) pair and focus instead on picking a policy that consistently yielded the high score. While this is an extreme example, we hope it clarifies why domain randomization is not guaranteed to yield robust behavior and do not see a resolution to the problem as a result of a more expressive policy. However, if the reviewer meant something else or if there are further questions and concerns, please let us know.
> >
> > Finally, we have slightly re-written the experiments section of the paper to clarify which tasks are in-distribution vs. out-of-distribution. Playing against tasks generated by our adversary class are in-distribution, whereas playing against mass and friction variations is out-of-distribution. Playing an adversary only provides guarantees on in-distribution performance and Figure 2 now shows that agents trained against a single adversary do not inherit this in-distribution performance guarantee for the reasons identified in the introduction. Additionally, we now test how domain randomization does on these in-distribution tasks and show that domain randomization does not provide robustness on these tasks either.
> > As the discussion of failure modes of domain randomization is interesting, but not the key contribution of the paper, we have moved the section formerly called "Hypothesis 3" into the appendix and direct readers there in the experiments section.
> > To summarize, we have made the following changes and additions (highlighted in red in the latest paper version):
> > - Compared the performance of agents across different adversary training regimes (agent trained against 1 adv vs. adversaries from the 3 adversary case, domain randomization agents against 1 and 3 adversaries, etc.). This is summarized in Fig. 2.
> > - We have added additional experiments on bandit tasks and a Deepmind control ball-in-cup task to demonstrate our method in more varied domains.
> > - We have rewritten the experiment section to indicate which tasks are in-distribution (for which we have lower bounded performance guarantees from adversarial training) vs. out-of-distribution (for which performance improvements are plausible but not guaranteed).
> > - We have moved the discussion on failure modes of domain randomization to the appendix and added an additional example of a failure mode of domain randomization.
> > - Edits to respond to particular writing issues / notation brought up by the reviewers.
> >
> > We hope that these changes address the main concerns brought up by the reviewers and are happy to respond to any further questions.

---

> > > ### Comment · AnonReviewer2 · 2020-11-24
> > > **Still unsolid motivation**
> > >
> > > Thanks for the response.
> > >
> > > However, a Nash equilibrium is only stronger. It's "both" minimax and maximin. In other words, my previous comment would still apply. The description in the paper that the learner would overfit to the adversary and another adversary could introduce further perturbation for the learner is simply wrong on the conceptual level. Furthermore, gradient updates are not guaranteed to find Nash equilibriums and solving bilevel optimization in general has much stronger convergence guarantees (since it still minimization or maximization) than finding equilibriums). Finally, the authors should discuss conditions for the existence of (pure) Nash equilibriums.
> > >
> > > The proposed technique from my perspective is more an optimization heuristic to finding the equilibrium, which is similar to the mixing idea used to find equilibriums in the optimization and the regret minimization literature. This paper needs a more solid theoretical foundation and review relevant literature mentioned above. I think the current handwavy interpretation is misleading to readers.
> > >
> > > Since this argument is used all over the paper, I decided to keep my original score.

---

> > > > ### Author Response · Authors · 2020-11-24
> > > > **Motivation is with respect to RARL**
> > > >
> > > > Dear reviewer,
> > > > Thank you for the comments. We would like to add the following clarifications to the discussion.
> > > >
> > > > > However, a Nash equilibrium is only stronger. It's "both" minimax and maximin. In other words, my previous comment would still apply. The description in the paper that the learner would overfit to the adversary and another adversary could introduce further perturbation for the learner is simply wrong on the conceptual level.
> > > >
> > > > The RARL (Pinto et. al.)  framework, which we are building on, explicitly is attempting to solve for Nash (which as we point out in our rebuttal, would have the same value if the solution set includes mixed equilibria). More directly though, since we are contrasting with this setting, we focus on Nash and use similar gradient based techniques rather than bi-level optimization.
> > > > >  In response to the point about “overfitting” being conceptually incorrect:
> > > >
> > > > A unimodal adversary will indeed cause the agent to “overfit” as shown in figure 2: at convergence agents trained against a single adversary **are not robust to adversaries from other seeds** even though those adversaries are technically in the training class.
> > > > For example, in a gridworld setting where the adversary places the goal state, a sharp unimodal representation would concentrate the goal placement into some particular quadrant of the world. Thus, at any moment in training the agent will have some belief about roughly where the goal should be hard-coded into its policy and will always attempt to go there first. This behavior would also consequently show up in testing settings. While we do not use the gridworld example directly in our paper, our robot vs. wind example has the same flavor.
> > > > > Furthermore, gradient updates are not guaranteed to find Nash equilibriums and solving bilevel optimization in general has much stronger convergence guarantees (since it still minimization or maximization) than finding equilibriums).
> > > >
> > > > Like most RL methods with function approximation in continuous settings, there is no guarantee of finding Nash and converging. This is in general a problem with the entire field and does not apply uniquely to our paper.  We think that work on constructing convergence guarantees is very interesting and worth pursuing and intend to study convergence analysis of population based methods in future work. However, we do not think the absence of convergence guarantees weakens the point we raise about overfitting that we explicitly demonstrate in Figure 2 and supported elsewhere by our experimental results.
> > > >
> > > > > The proposed technique from my perspective is more an optimization heuristic to finding the equilibrium, which is similar to the mixing idea used to find equilibriums in the optimization and the regret minimization literature.
> > > >
> > > > Our technique is explicitly an optimization heuristic that performs well in these settings; we do not claim otherwise in our paper and try to be clear about this. However, this optimization heuristic performs well and would be a stronger baseline in future works than the current baseline of single-adversary RARL that appears in the literature. As we demonstrate, there is no additional computational cost to doing so and significant benefit since single adversary RARL (Fig. 2) does not at all achieve its stated goal.

---

> > > > > ### Comment · AnonReviewer2 · 2020-11-25
> > > > > **RE: Motivation is with respect to RARL**
> > > > >
> > > > > Thanks for the response.
> > > > >
> > > > > I totally agree with the effectiveness of the proposed approach in practice, which provides a better quality approximation to equilibriums (either maximin or Nash) in terms of numerical optimization results.  My concern was about the current overall problem formulation is misleading, where the authors confuse the ideal solution concept that the algorithm aims to achieve and its properties, with what can be realized in practice due to optimization difficulties in using gradient updates to solve equilibrium problems. This paper addresses the latter but incorrectly interprets the former in motivation and explanations throughout the paper. As fixing this would constitute a major revision of the paper's argument, I intend to keep my current decision.

---

> > > > > > ### Author Response · Authors · 2020-11-25
> > > > > > **Solving for Nash**
> > > > > >
> > > > > > Thanks for the response. I think we are still unclear on the precise nature of the reviewers objection. It seems that the reviewer agrees that solving for Nash is an acceptable goal here.
> > > > > >
> > > > > > Would it clear things up in the paper if we stated more clearly that there are no guarantees (in some classes of problems) in getting a Nash equilibrium with gradient based methods and that we only are attempting (but not necessarily succeeding) in solving for Nash? We agree that naive gradient descent based methods have no guarantees of getting to Nash. Despite this, using gradient based methods in MARL systems is a standard approach despite the lack of theoretical guarantees. The RARL paper we are contrasting against is trying to solve for Nash and is at least finding a local equilibrium (since training appears to converge) and claiming that this local equilibrium (or the cycle around a local equilibrium in case we don't quite converge) is reasonably close in value to the global equilibrium. We are pointing out that this is not the case, that these local equilibria or cycles are extremely sub-optimal. Would it be helpful to just state that using this solution method is likely to lead to a local equilibrium or cycling behavior due to the use of gradient descent?
> > > > > >
> > > > > > We are trying to understand if the objection is to the language in the paper, to the solution concept, or to the actual underlying algorithm. The above response reads to us like an issue of language that we are happy to amend to be clearer around the theoretical constraints of these approaches. If we've misunderstood, additional clarity would be really useful in helping us to write an improved paper.

---

### Official Review · AnonReviewer3 · 2020-10-30
**The conceptual novelty is quite incremental but the experimental results are solid.**

**Rating:** 5
**Confidence:** 3

**Review:**

This paper extends the existing work on robust adversarial RL by training multiple adversarial agents from a population. Solid experimental results are presented to show that the proposed method improves the single adversary setting and domain randomization.

The experimental results in this paper seem solid to me. My biggest concern for this paper is that the conceptual novelty seems quite incremental. I mean, on the conceptual level, it is a common sense from robust control that multiple uncertainty sources can be treated together (e.g. the robust control theory handles the structured uncertainty in such a manner). One can augment all the adversary networks as a big network and then the problem formulation is the same as before. If we think each phi_i is a block in this big "single adversary network", then what the authors have done can be thought as doing  block coordinate descent. From this perspective, there is not too much conceptual novelty, and the main contribution of this paper is doing some more detailed study showing how to combine augmentation and adversarial RL. I am not sure whether such a contribution itself is enough for ICLR or not.

---

> ### Author Response · Authors · 2020-11-16
> **Main contribution is pointing out flaws in an existing framework**
>
> We thank the reviewer for their thoughtful feedback and constructive comments and are happy the reviewer found our experimental results solid. We agree with the reviewer that the conceptual novelty of this paper is limited but would like to clarify that conceptual novelty is not the main contribution of our paper. **In our view, the main contribution of this work is pointing out serious limitations in the use of a single adversary in continuous control tasks where the parametrization of the adversary is unimodal.** This parametrization is both used in the original, influential Robust Adversarial Reinforcement Learning framework and also appears as a baseline in many works (see [1], [2] as examples). As the reviewer points out, this is in contrast to the case in robust control where an entire class of structured perturbations is simultaneously considered. We view our work as explicitly drawing this connection and pointing out that constructing an agent that is robust against a single perturbation from the class is not the correct notion of robustness.
>
> Within the Robust Adversarial Reinforcement Learning framework and related frameworks, it is common to use a single adversary and as both Fig. 2 and our exposition point out, this does not actually yield robustness to perturbations that are within the class of training perturbations. Our contribution is noting the absence of robustness from a single uni-modal adversary and proposing a simple solution that leads to robustness in the domains that we examine. To further this point, the rebuttal version that we have uploaded has additional experiments in the appendix (highlighted in red) demonstrating the need for multiple adversaries in additional, more diverse domains: catching a ball in a cup and a bernoulli bandit task. Seeing as the training mechanism that we are responding to, Robust Adversarial Reinforcement Learning, is an influential work in the robust RL literature, we think that pointing out key flaws in the approach and pointing out that they can be approximately fixed with simple mechanisms is a valuable contribution.
>
> > "One can augment all the adversary networks as a big network and then the problem formulation is the same as before. If we think each phi_i is a block in this big "single adversary network", then what the authors have done can be thought as doing block coordinate descent."
>
> The point the reviewer raises about population based approaches being equivalent to a larger neural network with a categorical variable selecting which subnetwork is currently activated is an interesting point. Essentially, our population based approach is intended to roughly approximate a multi-modal policy but a large network with smaller subnetworks that are activated by some criterion is another way of representing multi-modality. As we discuss in the future work section, we also think it would be valuable to explore direct multi-modal adversary policies rather than the population based approach. However, it can be difficult to train policies that appropriately randomize as naive learning dynamics can cause agents to cycle through pure policies even when the Nash equilibrium is mixed (see “Learning with opponent-learning awareness” by Foerster et al for examples and possible solutions). Since we did not want to add additional training techniques to induce our adversaries to adopt a mixed nash policy directly, we suggest population based training as a simple alternative.
>
> > “From this perspective, there is not too much conceptual novelty, and the main contribution of this paper is doing some more detailed study showing how to combine augmentation and adversarial RL.”
>
> We are unsure what is meant by “augmentation and adversarial RL” as we are not using augmentations in this work. Could the reviewer clarify this statement?
>
> Please let us know if you have any additional comments or concerns. Thank you!
>
> [1] Ma, Xiaobai, Katherine Driggs-Campbell, and Mykel J. Kochenderfer. "Improved robustness and safety for autonomous vehicle control with adversarial reinforcement learning." 2018 IEEE Intelligent Vehicles Symposium (IV). IEEE, 2018.
> [2] Kumar, Saurabh, et al. "One Solution is Not All You Need: Few-Shot Extrapolation via Structured MaxEnt RL." Advances in Neural Information Processing Systems 33 (2020).

---

> > ### Author Response · Authors · 2020-11-22
> > **Additional changes + summary**
> >
> > Finally, we have slightly re-written the experiments section of the paper to clarify which tasks are in-distribution vs. out-of-distribution. Playing against tasks generated by our adversary class are in-distribution, whereas playing against mass and friction variations is out-of-distribution. Playing an adversary only provides guarantees on in-distribution performance and Figure 2 now shows that agents trained against a single adversary do not inherit this in-distribution performance guarantee for the reasons identified in the introduction. Additionally, we now test how domain randomization does on these in-distribution tasks and show that domain randomization does not provide robustness on these tasks either.
> > As the discussion of failure modes of domain randomization is interesting, but not the key contribution of the paper, we have moved the section formerly called "Hypothesis 3" into the appendix and direct readers there in the experiments section.
> > To summarize, we have made the following changes and additions (highlighted in red in the latest paper version):
> > - Compared the performance of agents across different adversary training regimes (agent trained against 1 adv vs. adversaries from the 3 adversary case, domain randomization agents against 1 and 3 adversaries, etc.). This is summarized in Fig. 2.
> > - We have added additional experiments on bandit tasks and a Deepmind control ball-in-cup task to demonstrate our method in more varied domains.
> > - We have rewritten the experiment section to indicate which tasks are in-distribution (for which we have lower bounded performance guarantees from adversarial training) vs. out-of-distribution (for which performance improvements are plausible but not guaranteed).
> > - We have moved the discussion on failure modes of domain randomization to the appendix and added an additional example of a failure mode of domain randomization.
> > - Edits to respond to particular writing issues / notation brought up by the reviewers.
> >
> > We hope that these changes address the main concerns brought up by the reviewers and are happy to respond to any further questions.

---

### Author Response · Authors · 2020-11-22
**After first revision**

Dear reviewers,

We sincerely appreciate the time and effort you took to review our paper.

Since the second discussion phase will end soon, please let us know if you have any comments/concerns that we have not addressed to your satisfaction. We will be happy to clarify further and strengthen our paper.

Thank you very much!

Authors

---

### Decision · Program_Chairs · 2021-01-07
**Final Decision**

**Decision:**

Reject

**Comment:**

The paper studies reinforcement learning in the presence of (adversarial) perturbations in the underlying system dynamics. The main (novel) observation is that  agents trained against a single policy may overfit  to that policy and hence will lack robustness to new/unseen policies. The paper proposes a population-based augmentation to the Robust RL formulation in which a population of adversaries are randomly initialized and samples from during training. The authors seek to show that their method generalizes well to unseen policies at test time.

Most reviewers agree that the paper provides a range of solid experimental results (with in-distribution and out-of-distribution tasks) showing robustness and generalization of their methods on several robotics benchmarks while avoiding a ubiquitous domain randomization failure mode. However, all the reviewers (and myself) agree that some of the conceptual claims of the paper may not be precise. For example, some of the reviewers disagree with the authors on finding the (mixed) Nash equilibria. Such general claims are hard to validate (may not even be true) and need theoretical justification. Hence, it is not conceptually clear why using multiple adversaries would not suffer from the same limitations as in the single adversary case.  Also, in the discussion phase, the reviewers agreed that the results/claims of the paper (i.e. overfitting to a single adversary and the need for multiple adversaries) are very interesting, but at the same time need to be confirmed by more extensive experiments.

Indeed, if the above are addressed, the paper would make a strong contribution to the area of RL.